# A Comprehensive Survey of Energy-Efficient MAC and Routing Protocols for Underwater Wireless Sensor Networks

**Zahid Ullah Khan** [1,2,3,*], **Qiao Gang** [1,2,3], **Aman Muhammad** [1,2,3], **Muhammad Muzzammil** [1,2,3],
**Sajid Ullah Khan** [4], **Mohammed El Affendi** [5], **Gauhar Ali** [5], **Imdad Ullah** [6] and **Javed Khan** [7]

1   Acoustic Science and Technology Laboratory, Harbin Engineering University, Harbin 150001, China
2   Key Laboratory of Marine Information Acquisition and Security, Harbin Engineering University,
    Ministry of Industry and Information Technology, Harbin 150001, China
3   College of Underwater Acoustic Engineering, Harbin Engineering University, Harbin 150001, China
4   Department of Computer Science, IT University of Lakki Marwat, Lakki Marwat 28440, Pakistan
5   EIAS Data Science and Blockchain Lab, College of Computer and Information Sciences,
    Prince Sultan University, Riyadh 11586, Saudi Arabia
6   College of Computer Engineering and Sciences, Prince Sattam Bin Abdulaziz University,
    Al-Kharj 11942, Saudi Arabia
7   Department of Electrical Engineering, University of Science and Technology Bannu, Bannu 28100, Pakistan
*   Correspondence: engr.zahidkhan09@hrbeu.edu.cn

**Abstract:** Underwater wireless sensor networks (UWSNs) have become highly efficient in performing different operations in oceanic environments. Compared to terrestrial wireless sensor networks (TWSNs), MAC and routing protocols in UWSNs are prone to low bandwidth, low throughput, high energy consumption, and high propagation delay. UWSNs are located remotely and do not need to operate with any human involvement. In UWSNs, the majority of sensor batteries have limited energy and very difficult to replace. The uneven use of energy resources is one of the main problems for UWSNs, which reduce the lifetime of the network. Therefore, an energy-efficient MAC and routing techniques are required to address the aforementioned challenges. Several important research projects have been tried to realize this objective by designing energy-efficient MAC and routing protocols to improve efficient data packet routing from Tx anchor node to sensor Rx node. In this article, we concentrate on discussing about different energy-efficient MAC and routing protocols which are presently accessible for UWSNs, categorize both MAC and routing protocols with a new taxonomy, as well as provide a comparative discussion. Finally, we conclude by presenting various current problems and research difficulties for future research.

**Keywords:** energy-efficiency; MAC protocols; ALOHA; TDMA; FDMA; bio-inspired E2RPs; cluster based E2RPs; reinforcement learning based E2RPs

## 1. Introduction

The corporate community and academia have long collaborated in order to meet the sustainable development goals (SDGs) set forth by the united nations, which place an emphasis on sustainable underwater wireless sensor networks (UWSNs) communication technologies with an intent for the wellbeing of marine life, better energy efficiency, and data security [1–3]. Furthermore, the unique underwater communication techniques should enable upcoming applications in the industrial internet of things (IIoT) and UWSNs. Many research groups from academia are focusing on enhancing the performance of UWSNs, particularly through reliable and energy-efficient communication for the localization of nodes in a cooperative network below water. However, this work focuses on two SDGs, namely SDG 14 for life below water and SDG 17 for sustainable development of science and technology, by enhancing effective node energy consumption methodologies in UWSNs [3–5].

The exploration of marine resources and the monitoring of the marine habitat have recently increased with the development of UWSNs. Which can be either the ocean/sea floor or a river bed, that has become more critical than ever before, due to its various applications [6]. Developed UWSNs became a major source of ocean exploration in recent times with having different applications, such as oil fields, cable line networks monitoring, pollution detection etc. [7]. As there are a lot of application of UWSNs, which includes environmental monitoring, disaster prevention, detection of oil gas leaks, and pollution monitoring [8]. On the other hand, the UWSN humanitarian perspective can help prevent manmade and natural disasters, boost economic growth and save marine life [6]. Specifying the significance of these applications in recent researches, the investigation and development of energy-efficient UWSNs protocols has sparked a lot of interest among the research community [9]. The deployment of UWSNs for different application obviously requires power/energy source to operate, so we have to look into the possibilities of such systems/technologies that can optimize power consumption with no additional manufacturing/resource cost to be added. Therefore, it is important to analyze MAC protocol, because it is the MAC layer, which is responsible of accessing sensor Rx nodes and Tx anchor nodes to share information between them in a deployed UWSNs. In UWSNs MAC protocols operates on the top of physical layer, correcting data mistakes, framing packets, and data flow management on the physical layer. These are all important elements of any efficient MAC protocol. Energy efficiency is always considered by the MAC protocols, which is crucial for the life and efficient functioning of UWSNs [10,11]. New MAC arrangements that rely on their media outreach strategies are being explored to highlight the problems inherited as a physical property, which should be considered in developing the MAC protocol [12–14]. A number of research groups have examined the UWSNs challenges such as, limited bandwidth, propagation delay, doppler propagation and high-rise transmission loss [15–17]. The aforementioned challenges must be considered when designing any protocol for efficient energy consumption in UWSNs [18]. Energy efficiency in UWSNs refers to the measures for reducing energy consumption by using minimum given resources to deliver a specific level of functionality, which benefits the user by lowering the cost of smooth functioning. UWSNs are made up of small, self-contained identical nodes called sensors Rx nodes, that collect data on a variety of physical and environmental characteristics such as temperature, sound, and wait movement in diverse areas [19]. If the given task requires it, it will further analyze the data before sending it to Tx anchor nodes for acknowledgement. From where the data will be transferred to base stations. It is basically the base station that provide a connection to the physical environment, where the gathered data is processed, broken down, and displayed according to the application's requirements [9]. To reduce the use of power resources and extend the network lifetime path while delivering data, energy efficient routing methods are necessary. Data must be transferred between the sensor nodes and the base station, which necessitates the use of routing algorithms. Energy conservation is a critical concern for extending the network's lifespan. However, the routing protocols are classified based on the operation that is utilized to fulfil the functions of a UWSN. For example, proactive protocols where the data is transferred using a pre-defined route, i.e., low-energy adaptive clustering hierarchy (LEACH) and power efficient gathering in sensor information systems (PEGASIS) [20]. On the other side, UWSNs place a great priority on building energy efficient routing protocols. In order to gather data in underwater habitats with minimal energy use, a number of routing techniques have been devised and published in the literature. The routing techniques took the characteristics of the devices and the sensor Rx nodes specs into account. Reactive protocol where the rout is established on demand dynamically, i.e., (TEEN), ad hoc on-demand distance vector (AODV), dynamic source routing (DSR) [21]. Whereas hybrid protocol are the ones where all the routs are first originated and then enhanced at the sending time of data. This hybrid protocol retain the concept of both proactive and reactive protocols, i.e., [21].

Routing protocols that are opted for energy efficiency in a UWSNs mainly consists of four different architectures, i.e., 1D, 2D, 3D and 4D architecture. A one-dimensional (1D) UWSNs architecture is the one, in which the sensor nodes are self-contained. Every sensor node is an independent network in itself that is in charge of sensing, processing, and relaying data to the remote station [5,22,23]. This style of architecture is known for its simplicity, where a node can be a floating buoy that detects underwater characteristics or a floating buoy that can be deployed underwater for a particular period of time to collect data [24], and then float to the surface to broadcast the collected information to the remote stations. In 1D-UWSNs, the nodes can communicate utilizing acoustic, optical or radio frequency (RF). Additionally, the topology of 1D-UWSN is star topology, which means that transmission between the sensor nodes and the remote station is passed on via a single hop [25].

The architecture of 2D-UWSN is made from a cluster of sensor nodes placed underwater. It consists of a cluster head with a fixed position that does not changes with functionality or time in the network. Each cluster node collects data and transmits it to the anchor Tx nodes. The anchor Tx nodes in the UWSNs collects and passes data from all of its sensor nodes to the surface floating nodes. The 2D-UWSN communication is carried out in two different dimensions (i) A horizontal communication link connects each cluster member to its Tx anchor node (ii) A vertical communication link connects the Tx anchor node to the surface buoyant node [26]. Acoustic, optical, and RF communication is opted in 2D-UWSN depending on the type of application and the nature of the environment. The 2D-UWSN can be used in both time-critical and delay-tolerant applications [27–29].

In 3D-UWSN the sensors nodes are arranged in clusters formations that are mounted at varying depths underwater [30]. As Three communication situations are included in this design. (i) K-means clustering transmission of nodes at dissimilar levels, (ii) Maximum inter cluster transmission, and iii) Anchor buoyant node transmission. Optical, RF and acoustic, communication linkages can be used in all three situations of communication [31]. 4D-UWSN architecture is the combination of 1D-UWSN and 3D-UWSNs that creates a four-dimensional 4D-UWSN. The remotely operated underwater vehicles (ROVs) collects data from Tx anchor nodes and deliver it to a central station which makes up the mobile UWSNs [6,32]. Each underwater sensor node can be autonomous in transmitting data directly to the ROV, depending on how close it is to ROV. Sensors with large amounts of data and proximity to the ROV can use radio links, whereas sensors with small amounts of data and a long distance from the ROV can use acoustic links [33,34]. Table 1 shows the relevant surveys and their key contributions, scopes and limitations.

**Table 1.** Relevant surveys and their contributions, scopes, and limitations.

| Ref. No | Year | Key Contribution | Scope | Limitation |
|---------|------|------------------|-------|------------|
| [35] | 2016 | Provides a comprehensive review on routing protocols for UWASNs | All the routing protocols have been classified into different groups according to their characteristics and routing algorithms, such as the non-cross-layer design routing protocol, the traditional cross-layer design routing protocol, and the intelligent algorithm-based routing protocol. | Energy-efficiency was not the core focus of this work. |
| [36] | 2016 | Provides a comparative analysis of routing protocols based on node mobility for UWSNs | This article focuses on routing protocols that were based on node mobility with a focus on analytical performance of routing protocols. | Energy-efficiency was not the core focus of this work. |

**Table 1.** *Cont.*

| Ref. No | Year | Key Contribution | Scope | Limitation |
|---|---|---|---|---|
| [37] | 2017 | Provides a comprehensive survey on localization based and localization-free routing protocols. | Covers routing issues and in its associated protocols for UWSNs. | Energy-efficiency was not the core focus of this work. |
| [38] | 2019 | Covers research on two enabling technologies for underwater communication: (i) Acoustic communication (ii) Magneto inductive communication, their channel propagation characteristics, challenges, and proposals to overcome these challenges. | Provides a comprehensive survey on existing works related to physical layer in a network for underwater communication using acoustic and magneto inductive mediums of communication. | Energy-efficiency was not the core focus of this work. |
| [39] | 2019 | Provides a comprehensive overview of latest research projects and emerging topics in underwater communication with a comparative analysis of acoustics, optical and electromagnetic communication for UWSNs. | Highlights related issues of each enabling technology with future prospects and provides recommendations for next generation enabling technologies in UWSNs. | Energy-efficiency was not the core focus of this work. |
| [40] | 2020 | This work aims to provide a thematic taxonomy to classify existing literature on UWSNs. | Discusses various aspects of UWSNs, such as: simulation platforms, network elements, enabling technologies, routing protocols, security and its applications. | Reviews energy-efficient routing protocols for network layer only and does not review energy-efficient techniques in other network layers for UWSNs. |

Ahmed et al. in [36] their review article made an emphasized discussion on routing protocols with respect to node's mobility, which is influenced by factors including vector, depth, clustering, AUV performance, and the path that a sensor node should take. Khalid et al. in [37] had held a conversation about localization-based and localization-free routing protocols during which they also discussed some of the relevant issues pertaining to UWSNs. Li et al. in [35] classified their research into several categories, including the non-cross-layer design routing protocol, the traditional cross-layer design routing protocol, and the intelligent algorithm based routing protocol. Gupta et al. in [40] had worked on the concept of data collecting in the UWSN, its classification that is based on routing service parameters, as well as some of the existing difficulties encountered when gathering data. Communication in wireless sensor networks or more specifically we can say in UWSNs requires a lot of power to complete a simple task of establishing a communication live link. However, in the UWSNs it is not possible to reach out very frequently to the power sources that are keeping the system alive so that communication may happen between anchor TX and sensor Rx nodes. In addition, this battery source is drained due to the large amount of energy consumption by the processing unit of a node present in UWSNs. Therefore, we either have to design new processing unit that will have the capacity to perfume tasks using as much less processing energy so that it may help to retain the battery source power for a long time, or take the second option, which is to design and implement efficient protocols that will help in maximum efficient energy consumption on processor level. In addition, through this study we are trying to address the 2nd aforementioned scenario to make the system more energy efficient which was abandoned by all the aforementioned review articles.

This article is unique from other survey articles, because it gives a comprehensive overview of current development of the most recent ALOHA-based, TDMA-based and routing protocols from recent publications for UWSN by highlights the advantages and drawbacks of each. The purpose of this study is to assist research individuals/groups in overcoming the challenges, faced in designing a UWSNs MAC and routing protocol,

such as power/energy consumption, limited memory source, long and flexible distribution delays, bit error rate and availability of limited bandwidth. Figure 1, depicts an overview of this paper. A generic schematic diagram of UWSNs is shown in Figure 2.

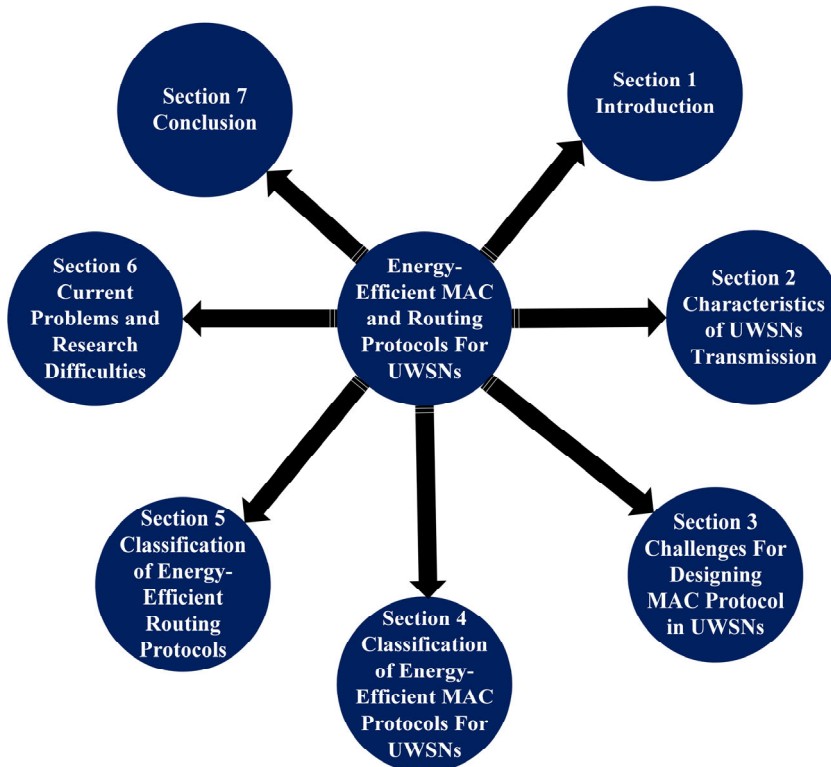

**Figure 1.** An overview of the paper.

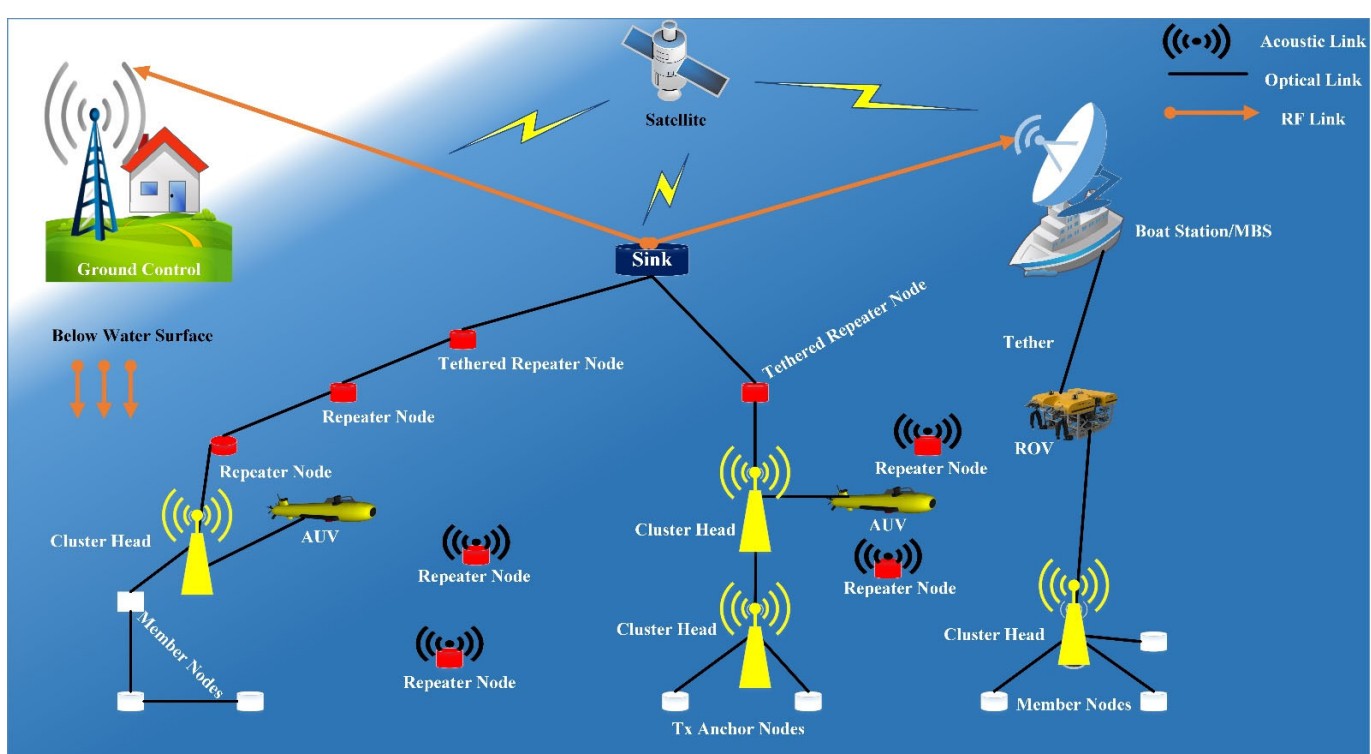

**Figure 2.** Scenario of underwater wireless sensor networks.

*Motivation and Contribution*

A developing network technology made up of sensor Rx nodes is mostly used for detection systems, submerged advanced warning, and aquatic environment element surveillance [28,33]. The connection quality of all these nodes are impacted by extra components including the Doppler frequency shift and ambient noise disturbance because of the submerged channels complexity [41]. The rate of communicating data packet transmission, the stability of transmitting data, network throughput, and energy usage of UWSNs are all directly impacted by these interferences. Consequently, it is a very difficult challenge to figure out how to send the collected information efficiently and quickly to the destination node. Data packet transmission from the Tx anchor node to the sensor Rx node in the network is guaranteed by MAC and routing protocols. The submerged habitat is complicated and unstable, and underwater sensor Rx nodes have relatively limited processing, memory, and communication resources. Due to the unique properties of UWSNs, including high power consumption, high transmission delay, and dynamic structure [42], the MAC and routing techniques offered for terrestrial wireless sensor networks cannot be simply adapted to underwater networks. Compared to TWSNs, the MAC and routing protocol of UWSNs is more complex and constrained. As a result, the submerged routing and MAC protocols should be able to provide very credible and efficient communication connections for the system in challenging underwater conditions. Scalable submerged routing and MAC protocols are required to support dynamic network updates and network stability during a wide range of circumstances. The route for transmitting data from submerged Tx anchor nodes to surface sensor Rx nodes is often chosen using routing and MAC protocols. Many publications about underwater MAC and routing protocols have been published recently as a result of an increase in the number of scholars who are interested in the study of underwater networks. The protocols for underwater routing and MAC were summarized in other papers. The majority of them, have worked on the secure data communication with having constant data rates in changing environment. However, in this study we focused on the efficient energy methodologies so to increase the life span of our opted co-operative networks. Abbreviations table, shows the abbreviations of our article.

The following are the major contributions of this work.

- Based on energy efficiency, we assessed several MAC and routing protocols in this study for which the process of sending and received data packets from source to destination is shown in Figure 3 for the better understanding of the readers.
- Figure 4 displays the RTS/CTS outcome for the carrier sense multiple access (CSMA) MAC protocol from the source to the destination communication in UWSNs with the Buffering_Slotted_ALOHA protocol network topology shown and elaborated through Figure 5.
- On the basis of routing techniques, we proposed a new classification of current E2RPs that is specially tailored for UWSNs.
- We explore the key ideas, guiding principles, benefits, and drawbacks of different proposed works, and offer a comparative analysis of routing algorithms with also providing a reliable cooperative routing strategies for UWSNs in terms of energy efficiency.

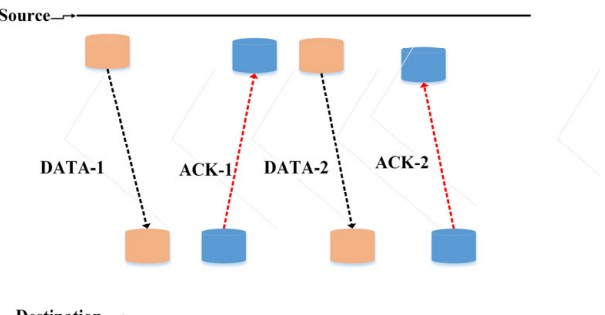

**Figure 3.** Procedure of sending and receiving data packets [43].

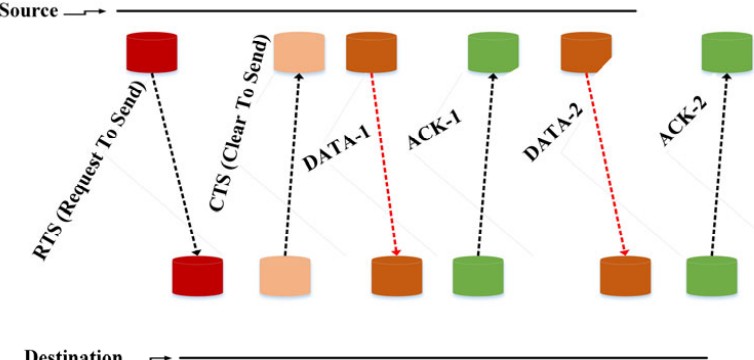

**Figure 4.** Result of RTS/CTS with CSMA [43].

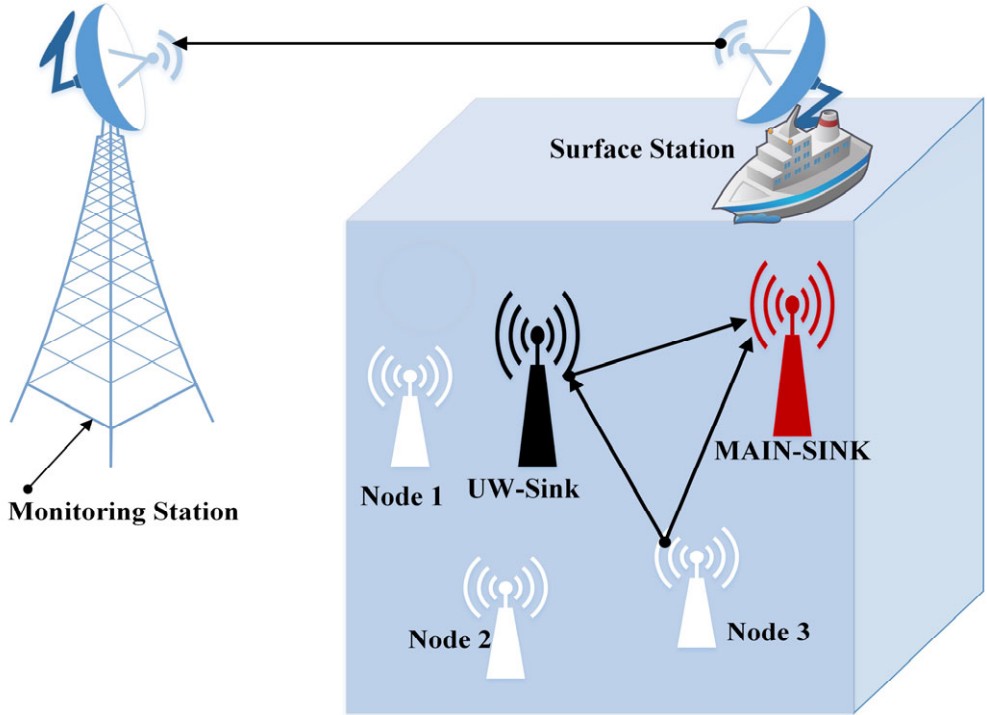

**Figure 5.** Buffring_Slotted_ALOHA protocol topology [44].

The remaining portions of this paper are listed below. In Section 1 provides a description of the introduction. Section 2 provides characteristics of UWSNs transmission. In Section 3 we discuss about the different challenges, while designing any MAC protocols in UWSNs. In Section 4 we categorized different MAC protocols in terms of their energy efficiency in UWSNs. In Section 5 we compared different routing protocols in terms of energy efficiency in UWSNs. The current problems and research difficulties pertinent to constructing an E2RPs for UWSNs are discussed in Section 6. Section 7 wraps up our research.

## 2. Characteristics of UWSNs Transmission

Data packet transmission between communicating nodes and addressing application requirements are basically the key objectives of UWSNs and TWSNs [13]. In an underwater habitat, radio waves attenuate rapidly [45]. Due to the aforementioned discrepancy the RF signals may travel very short distances. On the other hand, that phenomenon may differ in sea, fresh and river water, because they all have different level of visibility due to the presence of different materials in water, i.e., sand, dissolved minerals, mud, trash, animal feces, waste from different factories etc. For the aforementioned reasons, the optical

signal cannot propagate very far in UWSNs [46]. In comparison to optical and radio waves acoustical signals can travel extended intervals that is up to 1.5 km [47–50]. Due to this low attenuation feature of acoustical signals for long distances in comparison with optical and RF signals, underwater communication is mainly influenced by acoustical communication [51]. Different features of optical, acoustical and RF communication are given below in Table 2.

**Table 2.** Different features of various mediums in UWSNs.

| Characteristics | Optical Communication | Acoustical Communication | RF Communication | References |
|---|---|---|---|---|
| Bandwidth | From 10 to 150 (MHz) | ~1 Hz | ~1 kHz | [6] |
| Frequency Band | ~1014–1015 (Hz) | ~1 kHz | ~1 MHz | [9] |
| Speed of propagation. m/s | $3 \times 10^8$ m/s | $1.5 \times 10^3$ m/s | $3 \times 10^8$ m/s | [7,12] |
| Signal Attenuation | High | Low | Very High | [12] |
| The Size of Antenna | 0.1 m | 0.1 m | 0.5 m | [13] |
| Operational Range | From 10 m to 50 m | 1000 m | 10 m | [14] |
| Transmission Range | From 10 m to 100 m | 1500 m | 30 m | [14] |
| Attributes | Low Power Consumption High Data Rate Low Equipment Cost | High Power Consumption Medium Data Rate High Equipment Cost | High Power Consumption Medium Data Rate High Equipment Cost | [15] |

Nodes in a network needs to be active all the time, to maintain a flawless flow of data. Which is compulsory to complete different tasks in a network, weather TWSNs or UWSNs. A TWSNs network can benefit from a continuous power source, whereas UWSNs have limited power source. Therefore, high power consumption will shorten network life in UWSNs. Additionally, TWSNs are connected to global power supply networks, and are also equipped with solar power where the power sources can be changed frequently/easily, due to ease of access [16]. However, power sources in UWSNs cannot be changed frequently, because of the huge number of nodes spread in a network, as well as the harsh underwater habitats. This means with a limited power source in UWSNs will consequently shorten the life span of a network. For which we have to look into, density of nodes, integration of data, sleep time of a node, energy-saving algorithms and direction-finding regulations, which will directly extend the life of the UWSNs network [35]. We have written down the significant features to display the differences between TWSNs and UWSNs in Table 3. All of these features are significant factors in increasing network lifespan and improving network performance.

**Table 3.** Main differences among the characteristics of TWSNs and UWSNs.

| Features | UWSNs | TWSNs | References |
|---|---|---|---|
| Localization | GPS Non-Supportive | GPS Supportive | [17] |
| Stability of Links | Unstable | Stable | [23] |
| Transmission Range | Up To 2 km | 10–100 M | [27] |
| Transmission Speed | $1.5 \times 10^3$ m/s | $3 \times 10^8$ m/s | [30] |
| Energy Consumption | High | High | [33] |
| Data Rate | Low Data Rate | High Data Rate | [34] |
| Bandwidth | Limited | Limited | [48] |
| Bit Per Second Rates | Low | High | [49] |
| Transmission Delays | Extended and Flexible Transmission Delays | Small and Steady Transmission Delays | [49] |
| Noise | High-Influence | Low-Influence | [52] |
| Collective Association Technique | Acoustic Signals | Radio Signals | [53] |

## 3. Challenges for Designing MAC Protocols in UWSNs

UWSNs have their specific properties and structural restrictions such as, service restrictions, that consist of a restricted expanse of control, short range of communication, limited bandwidth, and restricted storage per sensor node [51]. The MAC protocols enable synchronized transmission between the sensor Rx node and Tx anchor node over a standard channel, regardless of the medium used for communication in UWSNs [46]. The underwater environment presents new problems that are needed to be considered during the development of the MAC protocol compared to the MAC intention for global networks [9,13,16,17]. Issues affecting the underwater communication are discussed in the sections ahead.

### 3.1. Restricted Bandwidth

UWSNs communication between nodes, depends on the medium (optical, acoustical, radio frequencies) that is opted to establish the communication link. Which are directly dependent on the available bandwidth of the propagating signal [25]. As the available bandwidth are limited in UWSNs, which directly casts off the transmitting data. UWSNs limited bandwidth can be a reason of network congestion, thereby data loss rates increase, which directly increase latency of the transmitted data packets between nodes. In addition, a delay in data transmission means that a system will keep running for long time to receive acknowledgements which in turn will increase power consumption [52,54].

### 3.2. Variabiity in Propogation Delays

As we already know that underwater speed of acoustical signals is $1.5 \times 10^3$ (m/s) and having propagation delay of 5 times more, when compared with the terrestrial radio frequency (RF) communication [53]. This high propagation delay in UWSNs is basically the combination of different factors, i.e., attenuation due to reflection/refraction of the acoustical signals from unpredicted seabed and the surface of water [55]. Temperature also plays an important role in making the propagation delays unpredictable/variable. Composition of the water is yet another factor that makes the delay variable. In addition, this variability in the propagation time between the sensor and anchor nodes makes our UWSNs accuracy poor, which directly affects the output that we were expecting to have. For instance localization of a sensor node, or sending some critical data to the anchor node so that it can broadcast that information to the base station [56]. All these factor that are making the propagation delay variable will directly impact on the formulation of MAC agreements [57].

### 3.3. Presence of Different Noise Sources

Noise exposure includes all man-made and natural noises. Natural noise refers to seismic and biological events which can cause noise in the environment, whereas man-made noise relates to noises from machines that work for different application. Due to these external noise sources in the communication channel, the connection between the sensor and anchor nodes may be disrupted [7,48,58].

### 3.4. Power Consumption

The devices that are used for underwater sound propagation, have a higher order of signal magnitude transmission capacity, than the devices that are used in terrestrial communication, which are equipped with advanced measurement of power transmission/consumption. Thus we will be needing specially designed protocols for acoustical communication to make it more power efficient [59]. Since UWSNs has many special features that make it unique to traditional networks. Power limitations is one example of the special feature, because once the power source of a sensor nodes drains off. It will require an instant replacement, as a delay in replacement will cause the communication between communicating nodes to suffer [52,55,59,60].

### 3.5. Doppler Spread

The change in the position of the transmitter or receiver is one of several causes of the doppler spread, which is also referred to as doppler diffusion in some studies [61,62]. In contrast to TWSNs, where RF is utilized as a medium of communication, this doppler shift is more evident in UWSNs due to the low velocity of acoustic signals used for communication between nodes [12,63]. It has been discovered that the doppler spread can be modified when the given bandwidth is limited [64]. The data rate of the entire communication channel will be reduced due to doppler shift, resulting in a decrease in the performance of acoustic communication system in UWSNs. As a result, when building an active MAC protocol for UWSN networks, it is critical to understand the fundamental properties of signal imbalances caused by doppler dispersion, which lowers the MAC protocols performance [38,65,66].

### 3.6. Synchronization

The difference between UWSNs and a TWSNs is considered to lie in the propagation delay and node movement. Therefore, to differentiate between nodes a time synchronization stamp algorithm will be required at MAC protocol design level for each sensor node in a UWSNs [67,68]. Due to the lack of accurate synchronization, the activity cycle method cannot ensure effective processing of sensory networks by dealing with the uncertainty of time between sensory nodes. This is due to the distribution delay factor is very high and changes from time to time [9]. The network underwater does not need to conduct global time synchronizations on a regular basis when employing time stamp synchronization, which cuts down the time it takes to synchronize clocks across sensor nodes [68].

### 3.7. Data Aggregation

Data aggregation in underwater communication is carried out by using three different mediums of communication (Acoustic, Optical and Magneto inductive) that varies with the necessity created with each specific application. Acoustics technology that is most feasible for long distance communication [69,70], However, optical way of communication is used for medium range of communication where a high data rate is required to perform the task at hand [71,72]. However, optical technology for data aggregation fails when there is abundance of organic material or any other kind of obstruction between anchor and sensor nodes. In such scenario where stopping communication is inevitable, MI communication is used for data aggregation [73,74]. In UWSNs, data aggregation is a critical method since, when UWSNs are deployed in remote areas or hazardous environments, data aggregation reduces energy usage by removing redundancy. The most difficult challenge with UWSNs is to extend their lifetime, which can be achieved with the help of data aggregation [7,14,55]. Where as to achieve data aggregation a central network is require shown at Figure 6 [75]. The main weakness of the central network is that it has one point of failure, means that one central anchor node is responsible for all the acknowledgments between the network of multiple nodes in water and base station on the ground. A central network will require a central channel too, which can be provided for a limited coverage area in UWSNs, but for large areas it is not the method of choice to be implemented [76].

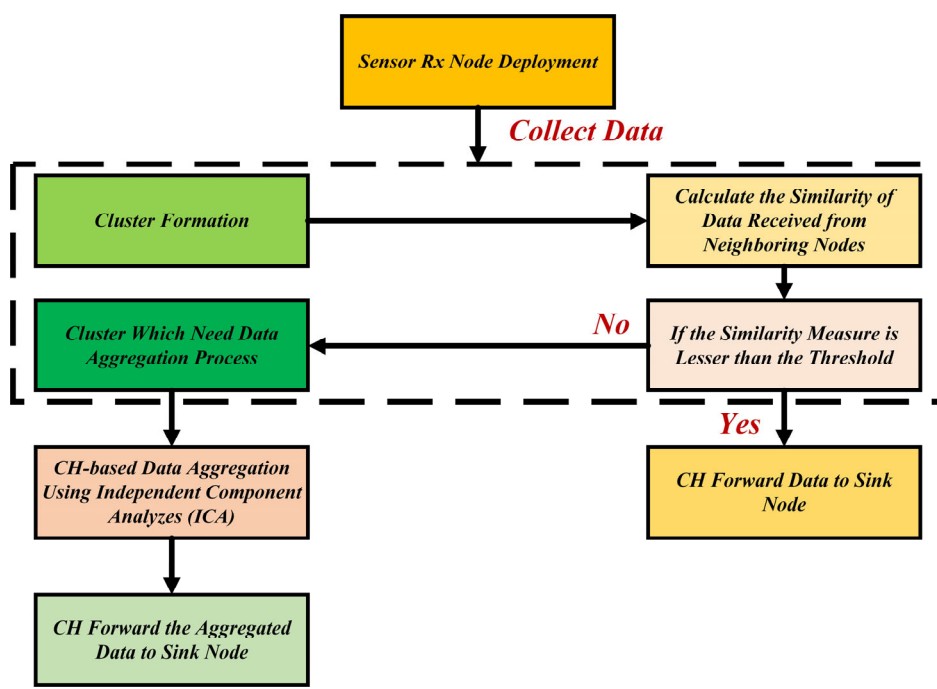

**Figure 6.** Flow diagram of data aggregation method [75].

## 4. Classification of Energy Efficient MAC Protocols for UWSNs

Here, we will discuss the MAC protocols classification for UWSNs and the developments in ALOHA protocols. The classification in Figure 7 has been divided into three major sections, i.e., frequency domain, full bandwidth and hybrid.

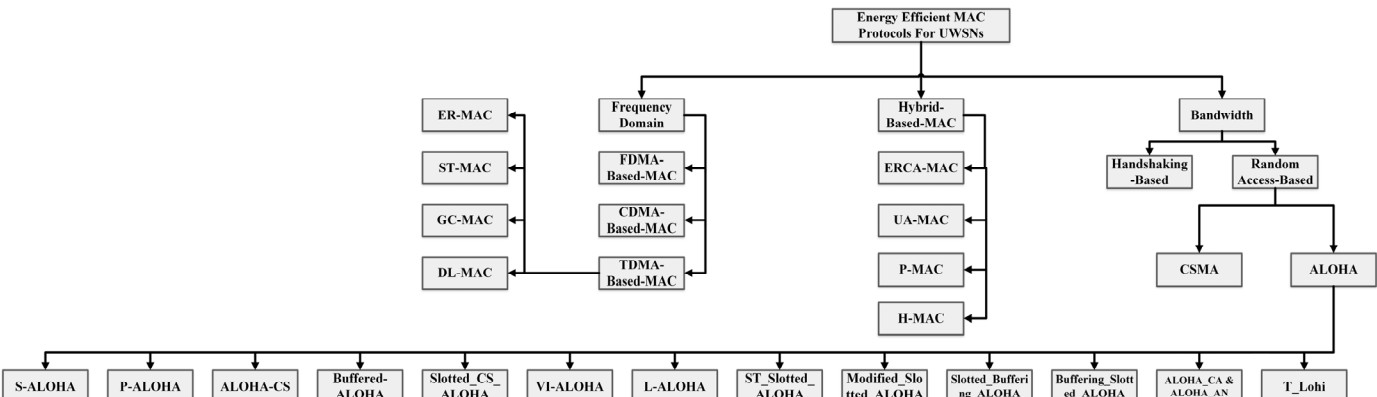

**Figure 7.** Classification of energy-efficient MAC protocols for UWSNs.

### 4.1. Frequency Domain

It is difficult to maintain the actual position between neighbor communicating nodes when considering the underwater acoustic channel's long latency. Simplified frequency domain protocol, which includes FDMA, TDMA and CDMA, is initially utilized in UWASNs. Simple and dependable FDMA divides the existing frequency band into many sub-bands and assign the sub-band directly into communicating node. However, a significant drawback is the availability of low bandwidth ratio [46,77]. In contrast to TDMA, CDMA protocols differentiate between users utilizing pseudo noise, which despite having a high channel usage and a straightforward technique, is still exposed to the "near-far effect" which is inherent to CDMA. The fundamental concept of FDMA is to select the appropriate frequency band for transmission in accordance with various transmission distances, which enhances the availability ratio of the bandwidth [78,79]. MAC protocols for the UWSNs

based on frequency domain, will be discussed in this section. Here we evaluate the MAC protocols frequency domain existing variations that are, multi-access technologies. Such as Frequency-Division-Multiple-Access (FDMA) Time-division multiple access (TDMA) and Code division multiple access (CDMA). MAC protocols for the UWSNs based on frequency domain, will be discussed in this section. Here we evaluate the MAC protocols frequency domain existing variations that are, multi-access technologies. Such as Frequency division multiple access (FDMA) Time division multiple access (TDMA) and Code division multiple-access (CDMA).

### 4.1.1. Frequency Division Multiple Access (FDMA)-Based MAC Protocol

The FDMA technique in UWSNs splits the available frequency band into sub frequency bands and assigns each sub frequency band to a single communication node, responsible for packet data exchange [64]. As a result, the bandwidth of FDMA sub channels used in between specific nodes in the network, is smaller than the overall bandwidth of the original transmission channel. This means that we can assign different bandwidth to same node in different time for different application but within the allowed limit of the total bandwidth of the whole channel [46]. Some FDMA-based MAC protocols proposed by different research groups are discussed in [80].

### 4.1.2. Code Division Multiple Access (CDMA)-Based MAC Protocol

CDMA allows numerous nodes in UWSNs to work within the same frequency band at almost the same time, and observing that signals from different nodes are identified using pseudo noise (PN) codes, which are essential to transport messages of different nodes in UWSNs [81]. The disturbances in communicating nodes are eliminated at the receiver's end that use dissemination procedures in order to receive the right message. CDMA technology allows high data transmission packets to be sent from one node to another at the same time. In [82] the authors have established a CDMA-based underwater MAC approach that allows for a periodic sleep mode which enables the communicating system to consume less energy.

### 4.1.3. Time Division Multiple Access (TDMA)-Based MAC Protocol

TDMA divides time intervals into several time slots (called frames), with each frame assigned to a particular communicating sensor node, where intervals and upper bits are merged into frames [64]. These time frames act as guards that play an important role to prevent the collisions of data packets in UWSNs, from adjacent sensor node, which is allotted a separate and unique time slot [83]. The simplicity of TDMA makes it an excellent choice for UWSNs at the MAC level, but the main drawback is the presence of propagation delay and delay variance in acoustic channels. In some scenarios, this makes synchronization between nodes difficult. Furthermore, designing the guard period is a vital component that must be addressed in order to avoid data collisions [64]. Ref. [48] provides a more detailed description of the shortcomings in TDMA.

*A.   Efficiency Reservation (ER-MAC)*

Efficiency Reservation MAC (ER-MAC) exploits the property of propagation time delay between communicating nodes, in a centralized topology, to establish the relative position of sensor nodes. It consists of cluster groups, the sink/anchor nodes in a cluster group is responsible of its member nodes utilization and reliably control of data transmission based on their relative location and directions. ER-MAC also implements duty cycling, where it incorporates the sleep mode into the communicating members of the cluster group to conserve energy [84].

*B.   Spatial Temporal (ST MAC)*

Spatial temporal MAC (ST-MAC) aspirations is to astound the spatial node environment in UWSNs. There is a great deal of uncertainty, by forming a spatial temporal-conflict graph due to large propagation delays in (ST-CG) [85]. Which is based on a comprehensive

graph coloring issue, that is resolved through the planned experimental TOTA. This allows the communication network to conserve energy while also increasing the throughput of the UWSNs GC-MAC, a novel energy-efficient, graph coloring-based UWSNs MAC protocol, which has been sprung up as a result of this technique [86].

### C. Graph Coloring MAC (GC-MAC)

Another energy-saving graph coloring MAC technique (GC-MAC) has just been suggested, similar to ST-MAC [87]. Due to the fact that ST-MAC uses a centralized scheduling method. It just necessitates a global understanding of the network nodes. Due to the high latency and limited data transfer rates in UWSNs, it is costly [88]. GC-MAC, in comparison to ST-MAC, can execute collision-free communication in a distributed manner without knowing the exact location of a node, which is required in terrestrial networking [8].

### D. Depth-Based Layering MAC (DL MAC)

In UWSNs, depth-based layering MAC (DL-MAC) provides energy-saving and collision-free characteristics. DL-MAC addresses near-far effects as well as hidden/exposed terminal concerns in underwater networks, in addition to spatial temporal uncertainty [89]. The suggested innovative protocol employs layering and a distributed clustering algorithm to efficiently communicate data packet exchanges, while reducing the likelihood of data collisions inside a network, hence improving the network's overall energy efficiency [90,91]. Table 4 depicts the summery of TDMA-based energy-efficient MAC protocols for UWSNs.

**Table 4.** Summary of TDMA-based energy-efficient MAC protocols for UWSNs.

| Protocol | Author/Year | Topology | Energy-Efficiency | Synchronization |
|----------|-------------|----------|-------------------|-----------------|
| ERMAC | (Nguyen/2008) | Centralized | Very High | Yes |
| ST-MAC | Hsu/2009 | Centralized | High | No |
| GC-MAC | Alfouzan/2019 | Distributed | Low | No |
| DL-MAC | Alfouzan/2019 | Distributed | Low | No |

### 4.2. Hybrid-Based Protocols

Hybrid protocols are another type of MAC classification that takes advantage of some of the frequency domain and bandwidth features [92]. Frequency domain protocols are more typically utilized for time-sensitive applications since they are more vulnerable to multipath hidden node issues and have larger and longer propagation delays [93,94].

#### 4.2.1. Energy-Efficient Reliable and Cluster-Based Adaptive MAC (ERCA-MAC)

A protocol called the energy efficient, reliable, and cluster-based adaptive MAC (ERCA-AC) was proposed in [69] to improve the network stability and helps in extending UWSNs life. This protocol separates networks into medium-sized clusters, based on the number of sensor nodes, to avoid collisions of data packets transmitted, to complete a task at hand. For communication between nodes in UWSNs, this protocol employs the TDMA mechanism, which makes it capable of dealing with the problem of hidden terminal nodes in a cluster of sensor nodes mentioned earlier [92]. This protocol also aids in making the networks throughput better and reducing propagation delay [13,92].

#### 4.2.2. Underwater Acoustic Multi-Channel MAC (UAMC-MAC)

In [93] researchers introduced the underwater acoustic multi-channel MAC (UAMC-MAC) protocol, which is based on the simultaneous usage of multiple channels, that is allowing communicating nodes to send data at the same time. This protocol combines CDMA with a handshake mechanism to accommodate a single hop's lengthy propagation delay factor and low throughput ratio between neighbors [94].

### 4.2.3. Preamble-MAC (P-MAC)

P-MAC is a hybrid protocol that combines a frequency domain MAC protocol with a slotted multiple access collision avoidance (Slotted-MACA) protocol. The inaccuracies caused by the loss of time synchronization can be solved with this protocol [95]. The default distance level information, which is a file containing evaluated and accumulated knowledge about the channel status and changes collected over periodic monitoring of the undersea environment, is used by P-MAC to operate dynamically and adaptively [96].

### 4.2.4. Hybrid-MAC (H-MAC)

Both frequency domain and random-access MAC techniques are used in the hybrid-MAC (H-MAC) protocol [46]. To deliver data in a collision-free way, this protocol divides the time frame into two time slots, with each communicating node using one of the time slots to perform the task at hand. The random access-based second portion of this hybrid protocol is utilized to adjust the changing traffic conditions for communication between nodes in UWSNs [97]. This adaptation of hybrid H-MAC makes it a very suitable candidate where use of energy in the UWSNs is a big constraint [17]. The key differences between the ERCA-MAC, UAMC-MAC, P-MAC and H-MAC methods are shown in Table 5.

**Table 5.** Cross correlation between UAMC-MAC, ERCA-MAC, P-MAC and H-MAC protocols.

| Protocol | Collision Rat | Network Topology | Simultaneous Transmission | Throughput | Power Consumption | Propagation Delay |
|---|---|---|---|---|---|---|
| UAMC-MAC | Medium | Ad-hoc, stationary | Yes, during one session | High | Medium | Low |
| ERCA-MAC | Low | Cluster, stationary | No | Medium | Low | Medium |
| P-MAC | High | Ad-hoc, stationary | Yes | High | High | Low |
| H-MAC | Medium | Ad-hoc, stationary | Yes | High | Low | Low |

### 4.3. Bandwidth

Based on geographical, spatial and temporal uncertainty, narrow bandwidth, near and distant field communication issues, time synchronization, and throughput performance variations. Frequency domain for MAC protocols is not the first choice of communication technique to be used for UWSNs [98]. The bandwidth MAC protocols, on the other hand, have access to the entire bandwidth of the connection channel and can distribute network resources on demand. As a result, the majority of the work on MAC protocols for UWSNs has been focused on bandwidth domain MAC protocols [99]. Table 6 shows, the significant differences between the frequency domain and bandwidth MAC protocol.

**Table 6.** The evaluation among frequency dominion and bandwidth MAC protocols.

| Factors | Frequency Dominion | Bandwidth |
|---|---|---|
| Scheduling | Central | Spread |
| Channel usage | Low | High |
| Network resource sharing | Reserved for a certain user | On demand |
| Appropriate network load | Low | High |
| Appropriate node density | Low | High |
| Appropriate network size | Small | Big |
| Ratio of collision | Low | High |
| Throughput | Low | High |
| Energy consumption | Low | High |
| Propagation delay | High | Low |

### 4.3.1. Handshaking-Based

Another significant type of bandwidth MAC protocol is the handshake protocol, which is basically a collection of reservation-based protocols [46]. The major goal of the handshake method is to avoid a collision of data packets, therefore before transmitting any data packets, the sender must check the channel state by sending request-to-send (RTS) and clear-to-send (CTS) control packets on the control channel. The handshaking can be performed in both single channel and multiple channel-based communication between nodes in UWSNs [9,100]. Before any payload is transferred over a single channel, channel handshake messages are exchanged [47,52]. A series of protocols targeted for promoting energy efficiency is one of the main themes of MAC protocols. multi-channel MAC protocols handshaking is distinct from single-channel MAC protocols handshaking [101].

### 4.3.2. Random Access-Based

In this method, the node begins broadcasting as soon as it has a packet ready to send. The data packet can be successfully received if the receiver is not busy and there is no conflict in the generated acknowledgements [102]. Using random access technologies, multiple nodes in UWSNs can share the transmitting medium in a random manner [46]. In the section of ALOHA protocol, we will go over this concept in greater depth.

*A.  Carrier sense multiple access (CSMA)*

The CSMA protocol is an analogous class of random-access protocols in which all nodes first sense the channel's attributes before using it for communication [103]. This prevents the user from wasting the limited communication resources available in UWSNs. A brief description of the aforementioned technique can be found in [13]. Although RTS/CTS-based protocols such as CSMA have outperformed ALOHA protocols in terrestrial networks. Their efficiency in UWSNs could be quite low due to the significant propagation delay discussed in [104]. CSMA based on RTS/CTS and procedure of sending and receiving data packets are depicted in Figures 3 and 4 [104].

*B.  ALOHA*

The ALOHA Protocol is a form of random-access protocol that capacitates numerous communication stations in UWSNs, to send data packets that may also be in the form of data frames, over the communication channels of same features and properties at the same time. The aforementioned protocol is an uncomplicated way of communication in which each network communicating stations are given equal priority, but functions independently [43]. The main differences between carrier sense multiple access (CSMA) and ALOHA Protocols are shown in Table 7.

**Table 7.** The main differences between CSMA and ALOHA protocols.

| Features | CSMA | ALOHA |
|---|---|---|
| Presentation in WSNs | Unchanging | Unchanging |
| Presentation in UWSNs | Not unchanging | Unchanging |
| Utilization of channel | High | Low |
| Limitations of optimization | The transporter intellect is starting point which is attuned | adjust the unpleasant back-off time |
| Energy ingesting | Low | High |
| Rate of Collision | Low | High |
| Transmission delay | Actual high in Underwater | High |
| Left over nodes of the network | Average | Lesser |

*(i)  Slotted-ALOHA (S-ALOHA)*

The slotted-ALOHA (S-ALOAHA) protocol was developed to increase the performance of pure ALOHA by preventing data packet collisions during communication between nodes in a UWSNs. The whole time window is broken into smaller sub time slots in

order to avoided data collision [105,106]. With this sub time slotting method, each node must send a data packet in its assigned time slot otherwise, it must wait for all other nodes to deliver their data in their allotted sub time slots. During transmission and acknowledgements, this approach ensures that no data packets clash [107]. The (S-ALOHA) approach will take longer to accomplish a task as a result of the preferred way of sub item slotting [82,106].

*(ii)    Pure-ALOHA (P-ALOHA)*

If we increase the size of the sub time slots discussed earlier in (S_ALOHA), it will be converted into pure-ALOHA (P_ALOHA). When P-Aloha is studied for communication in UWSNs, the performance is similar to that of RF networks since P-Aloha does not have a collision control system and frame collision is random [108]. Both these properties made the performance of the P-ALOHA poor as it will drastically drops the throughput of the system. The end-to-end delay will also be increased. In addition, as the control system works randomly in P-ALOHA, it will take more time to receive and send acknowledgements about data packets. Which in turn will make the system to use more energy [109].

*(iii)    ALOHA with carrier sense (ALOHA-CS)*

The ALOHA-CS protocol simply observe the network that weather its half-duplex node is receiving any data packets from any node that is trying to communicate with it in a UWSNs. Due to which the sensor nodes would never send any new data packets while listening for a data packet within the network, regardless of whether it is the intended recipient or not [110,111]. In the underwater acoustic environment, this protocol has taken advantage of a long propagation delay. Furthermore, when compared to pure ALOHA, this protocol delivers a significant increase in network throughput ratio, where the data packet size is large and the network has few nodes. Otherwise, throughput rapidly declines.

*(iv)    ALOHA with advance notification (ALOHA-AN)*

ALOHA-AN is based on a concept similar to ALOHA with collision avoidance (ALOHA-CA). The goal of ALOHA-CA is to overcome the limitations of ALOHA-CS, while it is widely acknowledged that ALOHA-CS has the advantage of preventing data packets from being sent, when listening to another sensor node in the same network [110]. Furthermore, the listening procedure carried out by a sensor node can occasionally be assisted in reducing the likelihood of collisions of data packets [110]. ALOHA-AN has to collect and store more data, which necessitates the use of resources other than ALOHA-CA [112]. When the packet size is big and there are few nodes in the network, this protocol gives a significant increase in network throughput compared to pure ALOHA.

*(v)    Buffered ALOHA protocol*

Several research groups have used the buffers to improve the ALOHA protocol's performance [113,114]. In [115] the authors developed an approximation approach for analyzing the S-ALOHA technique, which is based on a small user group with restricted storage capacity. The assumption of channel asymmetry was the main proof of concept of their method. The performance of arbitrarily selected users, which they refer to as tagged users, that are used for the analyses the system's performance. Authors in [116] had studied the implementation of S-ALOHA using smaller number of packet buffers. While the behavior of the hybrid ALOHA/TDMA protocol with client-side buffers had been investigated in depth by authors in [117]. The authors of [118] proposed the Buffered ALOHA, and had also investigated the influence of buffering packets on P-ALOHA. The authors of this paper presented a formula for calculating ALOHA throughput for a given number of active nodes. They divide the causes of failure into three categories on the basis of dropped data packets [118].

*(vi)    Slotted carrier sense ALOHA (Slotted_CS_ALOHA)*

The authors in [99] highlighted the issue of energy efficiency in UWSNs. For that they had presented the slotted carrier sense ALOHA protocol (Slotted_CS_ALOHA) as a

solution. Furthermore, if a packet collides while being sent from source to destination, it must be resent to receive a successful acknowledgement of the packet, which consumes energy. As a result, the sensor node will become ineffective and stop serving after a short period of time, and the data in this field will be wasted. Therefore, the authors in [119] address this issue by introducing a sleep mode whenever a node is not sending data and is inactive, to reduce power consumption. Prior of going into sleep mode, the node was allowed to send more than one packet. These data packets would then be sent to the buffer where the two conditions, slot time and CS would be used to reduce the possibility of collusion. In addition, there is a second buffer that will handle the ALOHA cycle, feeding only one packet to the system at a time. In addition, when that packet successfully reaches its destination, it will send another packet, and so on. It is found by the authors of the aforementioned study that while this strategy reduced process power consumption, and increase throughputs. It did not improve average delay.

*(vii) Variable interval ALOHA (VI-ALOHA)*

This protocol is specifically designed to reduce the likelihood of data packet collisions with one another within a network. It divides the broadcast channel used by anchor nodes, into sub channels, hence lengthening the time slot indirectly. For which the authors of [120] developed a variable interval ALOHA (VI-ALOHA) protocol with a randomly changing interval time slot and compared it to equal interval ALOHA (EI-ALOHA). To demonstrate the effect of the two protocols, that how they can reduce collision by increasing randomization in space. Where a variable interval was used to reduce the intersection of a beacon coverage. Secondly, they employee the position random distribution approach to generate a random beacon interval, which increases the randomness of each beacon broadcast, while reducing collisions caused by equal intervals.

*(viii) Learning-ALOHA (L-ALOHA)*

According to the authors in [121], two aspects are needed to be taken care off when implementing the learning-ALOHA (L-ALOHA). Firstly, it takes care of the learning algorithm that will be used while communicating between nodes, in which the node sends data packets at random intervals to identify a successful acknowledgement, where no collision between data packets occurs, which in turn will help to avoid data retransmission. The second portion is the steady component, which occurs when the entire network's learning process is stable. Where each node has ready data packets to transmit over the network and simply needs to be able to deliver data packets at a predetermined time period. The authors of the aforementioned research only compare their approach to S-ALOHA and P-ALOHA, where just two metrics (throughput and average end-to-end delay) are compared to demonstrate the magnitude of the differences.

*(ix) Saving time slotted carrier sense ALOHA (ST-Slotted–CS_ALOHA)*

Saving time slotted carrier sense ALOHA protocol is regarded as an update to Slotted_CS_ALOHA protocol. Which was designed and analyzed by the authors in [122]. This protocol uses one buffer, to allow the communicating nodes to generate and send numerous data packets, while also altering the position of other buffers to allow data packets to be routed back to their intended destination if they collide. The ST-Slotted–CS_ALOHA protocol outperforms Slotted_CS_ALOHA in terms of improving the energy consumption of the network, increasing throughput, and lowering the average delay factor, but the number of communicating nodes drops drastically, when compared to the Slotted_CS_ALOHA protocol.

*(x) Modified-Slotted-ALOHA*

The modified-slotted-ALOHA protocol is a recommended protocol for resolving issues, when an acknowledgement is not received [123]. This protocol employs a modulated buffer to help create many data packets and resend them to a communicating node in UWSNs. This modulation scheme will save energy that would otherwise be wasted by the sensor nodes. This protocol also overcomes the problems of low throughput, and high

average delay factor by employing a buffer, which saves data packets before sending them. Therefore, they can be resent if a collision happens, or an acknowledgement is not received. This will increase the average data transfer rate and solve the power consumption issue. The proposed protocol relies on back-off technology, which uses random time to determine the best time to transmit the data packets in UWSNs [123].

*(xi)   Slotted-Buffering-ALOHA*

The Slotted-Buffering-ALOHA protocol had been used to save energy and increase the lifetime of networks by researcher in [123]. They mentioned that when it comes to energy efficiency, there is a trade-off with delay in time. This protocol is designed to fix issues that have been identified in previous sections. More conditions are employed in this protocol to ensure that a collision does not occur. The first condition of which is to prevent the node from sending any data packets before the time window begins. The alternate option is to employ carrier sensing (CS) to send abbreviated messages over the control channel to characterize the state of the connection channel. Ensuring that no data is present in the communication channel and thus no collision or loss of data packet happens. Another important feature of this protocol is the usage of buffers to aid in the creation of numerous data packets.

*(xii)   Buffering_Slotted_ALOHA*

Buffering_Slotted_ALOHA discussed and analyzed in [44] is a new protocol for dealing with typical difficulties at UWSNs [124]. It is employed to reduce the migration of nodes from one group to another. This protocol divides the accessible network into discrete parts called closed groups. Every closed group within the accessible network has a tiny pool with a predetermined number of nodes for lowering traffic within the group, avoiding collisions, and thus reducing sending time. Another important feature of the closed group is the presence of an underwater sink (UW-Sink) node, which functions as a leader and interacts with a small number of communicating nodes within the closed group. Outside the closed groups, the Underwater Main Sink (UW-Main Sink) is responsible for data arriving from UW-Sink or normal nodes within the closed groups. As illustrated in Figure 5. The main purpose of a time slot is to allow each communicating node in a closed group to deal with the UW-Sink as a default choice if it is available, or with the UW-Main Sink as a backup option if the default choice is not available [125]. Furthermore, until the sending data packet is validated, it is retained in a buffer in each node. Checking the time-slots cases before sending a data packet is a key goal for speeding up the communicating process. The topology of this network is also shown in Figure 5. It is found out that this protocol can boost network throughput while lowering the average end-to-end delay factor and energy consumption ratio. Different types of ALOHA protocols are compared in Figures 8–11 based on energy consumption, average propagation latency, throughput, and dropped nodes.

*(xiii)  ALOHA by collision avoidance and ALOHA by prior notification (Aloha-CA, Aloha-AN)*

The authors of the [110] have proposed two different types of ALOHA protocol, i.e., ALOHA by conflict avoidance (ALOHA-CA), and ALOHA by prior notification (ALOHA-AN). These both types are making the communicating system being able to increase the efficiency and lower the energy consumption by decreasing the amount of collisions between data packets in a UWSNs. To avoiding conflicts in ALOHA-CA, each node monitors the channel's condition, so that to keep a check on the noisy moment generated by a given frame of the channel's independent alternatives. This process is upgraded to ALOHA-AN by adding a transmission node, that has the capability to sends an acknowledgement to its neighbors prior to data transfer between nodes. The ALOHA-AN is resource intensive as compared to ALOHA-CA.

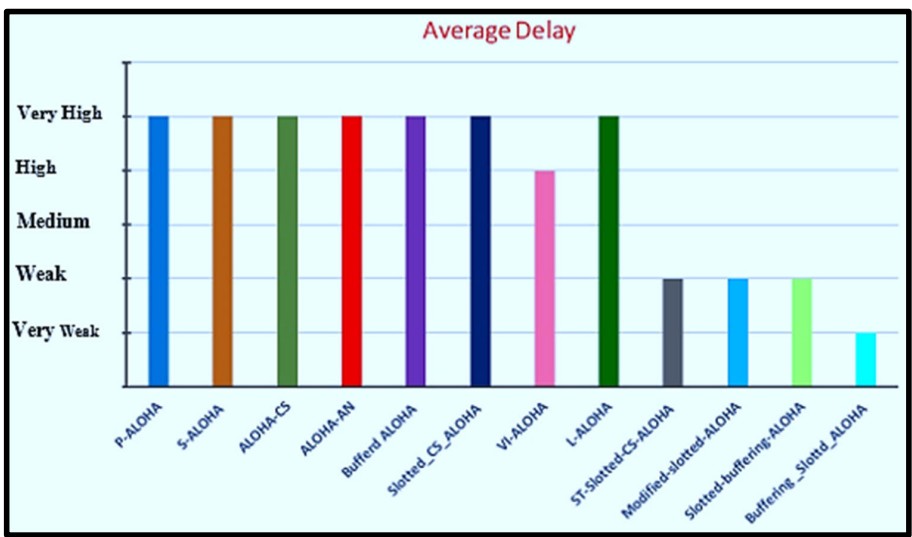

**Figure 8.** Comparison of average delay between different ALOHA protocols.

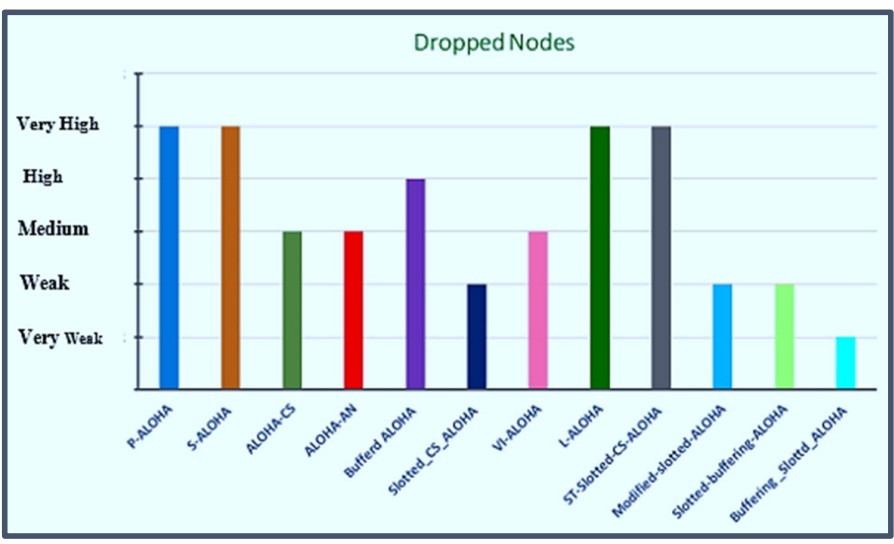

**Figure 9.** Comparison of dropped nodes between different ALOHA protocols.

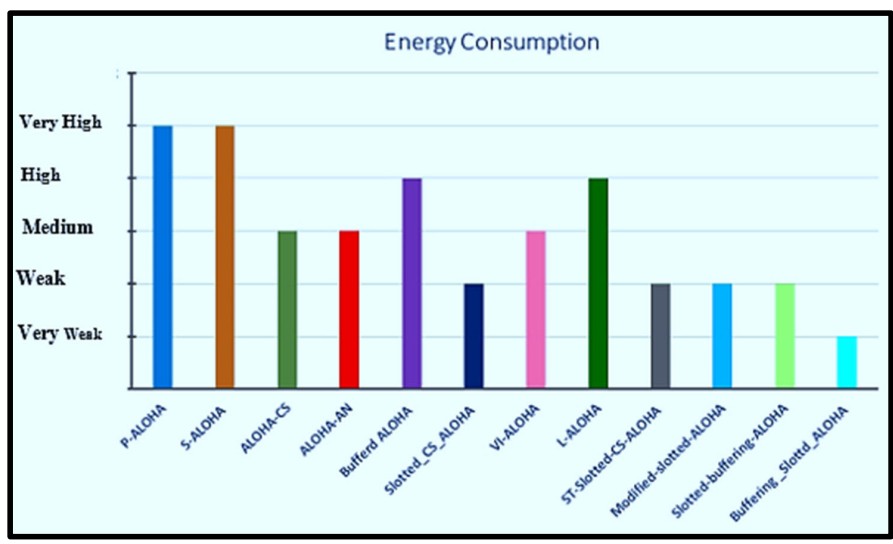

**Figure 10.** Comparison of energy consumption between different ALOHA protocols.

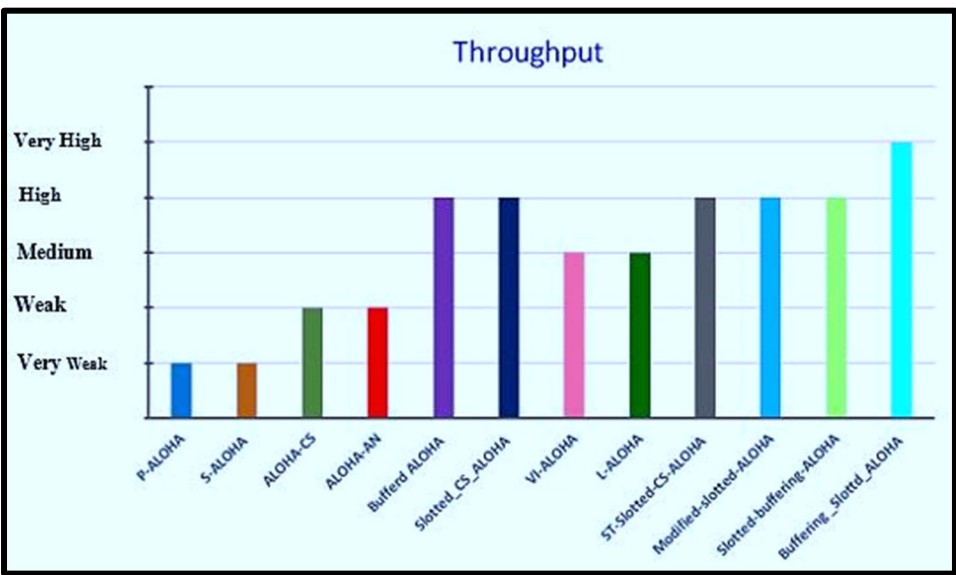

**Figure 11.** Comparison of throughput between different ALOHA protocols.

*(xiv)  Tone-Lohi (T-Lohi)*

Another MAC energy-efficient protocol is created on a tone based booking technique [53]. The sender node communicates a small tone as well as hears the channel during a dispute cycle. The transmitter channel is reserved in case if other communicating tones are not present. However, if there an alternative tone accrues, the backs up will try to relater the message so that it can be understood by the receiving node [126].

## 5. Energy-Efficient Routing Protocols (E2RPs) for UWSNs

The network layer attempts to enable cooperative sensing, connection, and data packet routing between the sensors and communicators. However, the routing protocols must be created to meet the necessary performance criteria in UWSNs in order to promote energy efficiency. Convergence, robustness, and scalability are some of these requirements. The main goals of all these routing methods in UWSNs are to increase network lifetime and supply communicating nodes in efficient and reliable ways. The main factors affecting a routing protocol's energy efficiency in UWSNs are computing costs, communication, and neighborhood discovery.

We offer an original categorization of E2RPs for UWSNs in Figure 12. Current UWSN E2RPs fall into five categories: (i) bio-inspired, (ii) cluster-based, (iii) reinforcement learning-based (RL), (iv) cooperative reliability-based, and (v) depth-based. The bulk of E2RPs used in UWSNs uses routing protocols, based on cluster and cooperative reliability. However, due to the hasty advancement in routing protocols, which are based on artificial intelligence, bio-inspired and reinforcement learning are attracting a lot of attention. This is because they can choose the best routing protocols and are able to quickly adapt to the changing UWSNs environment. The existing E2RPs for UWSNs are thoroughly reviewed in this section in terms of their basic operating principles.

### 5.1. Bio-Inspired Energy-Efficient Routing Protocols (Bio-Inspired E2RPs)

Different technological advancements have been influenced by biological concepts across a variety of scientific areas [127]. The listed below protocols are reviewed in terms of E2RPs for UWSNs.

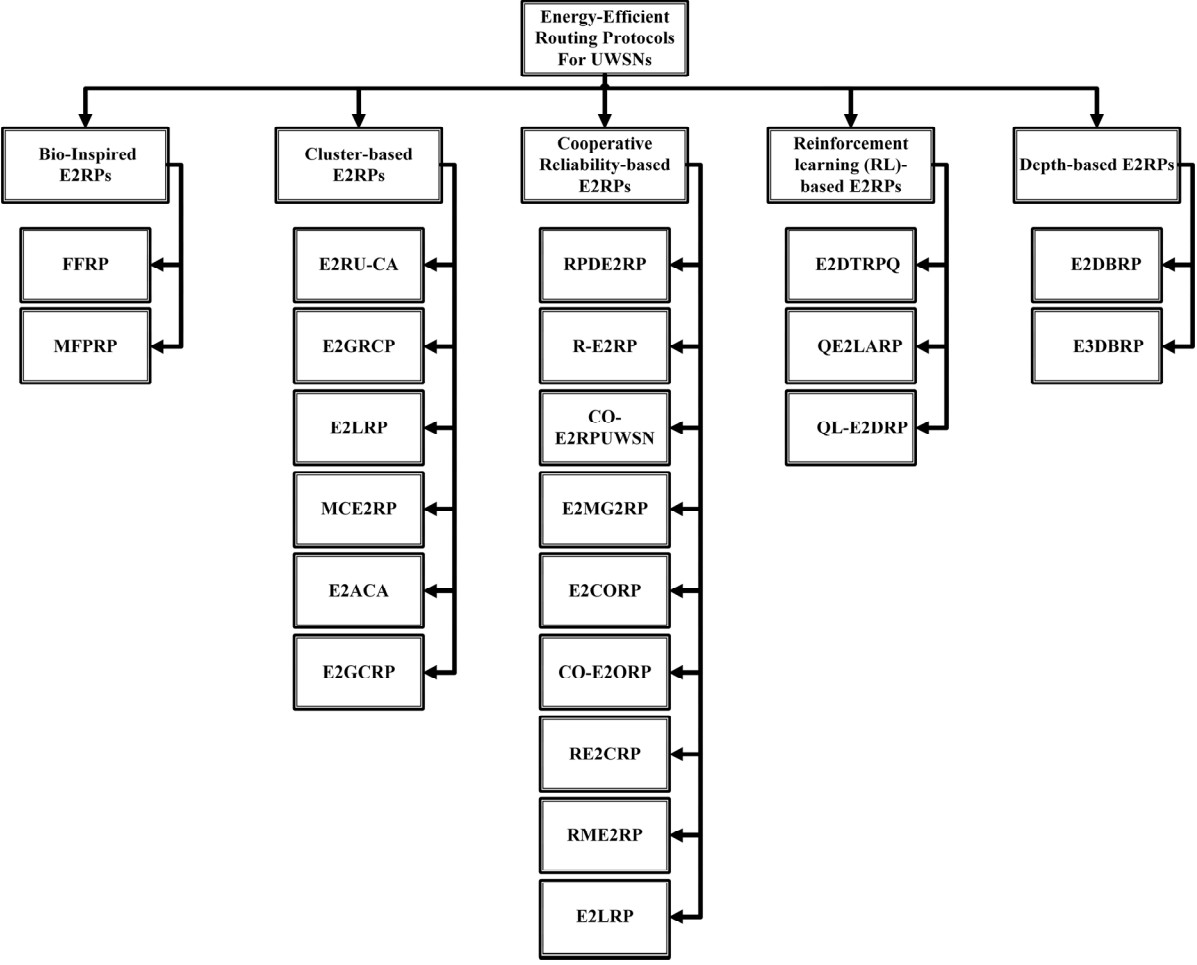

**Figure 12.** Classification of energy-efficient routing protocols.

### 5.1.1. Firefly Mating Optimization Routing Protocol (FFRP)

The authors of [128] presented the firefly mating optimization routing protocol (FFRP), which is dynamic and bio-inspired. In the process of optimizing firefly mating, pheromones play a vital role, which are generated from the body. There are two different kinds of fireflies: female fireflies and male fireflies. The leading Tx anchor node is selected by the firefly mating optimization routing protocol based on the highest value of these two kinds of fireflies. The maximal value of FFRP depends on the location of the communicating node in relation to the distance of the neighboring nodes, angle of departure, remaining energy, and water. The overflow duration of a buffer is a special parameter that reduces buffer overflow in communicating nodes. The FFRP uses the amount of remaining energy and the proportion of properly transmitted data packets over the connection to determine the connection quality. With the growth in data speeds, this technology assures robust and consistent connections. The simulation results are software based, as well as the densities of various numbers of nodes in the network in comparison with other routing protocols. As an example, see MERP in [129] and QERP in [130]. The simulation results demonstrate that, when compared to other current routing protocols, energy efficiency, network performance, and data packet delivery ratio are all improved by the FFRP.

### 5.1.2. Memetic Flower and Energy-Efficient Pollination Routing Protocol (MFE2PRP)

In [131], the Memetic flower and energy-efficient pollination routing protocol (MFPRP) was introduced to improve network quality of service (QoS). The proposed protocol's major goals are to select the route that maximizes data packet ratio while minimizing end-to-end latency in UWSNs. The MFE2PRP fitness value enhances the network's ability to convey

communicating data packets through secure connections with little energy consumption. The suggested routing system in [131] was evaluated and compared with other routing protocols, including the balanced multi-objective optimized opportunistic routing protocol (BMOORP) and the QoS aware evolutionary routing protocol (QERP). The suggested routing system achieves successful outcomes in terms of data packet delivery ratio due to the selection of optimum links for data packet communication.

### 5.1.3. Comparison of Bio-Inspired Energy-Efficient Routing Protocols

The properties of the physiologically affected optimum routing protocols are compared in Table 8 and are covered in this section. The construction of the best, most appropriate algorithms can benefit from biological insect modification to meet the many self-configuration and self-organizing problems in UWSNs. Furthermore, in terms of energy efficiency, network throughput, and end-to-end latency of the network, bio-inspired routing algorithms are thought to be more effective routing protocols for large-scale UWSNs. Consequently, the vast majority of bio-inspired routing protocols are quite effective. Therefore, these routing methods can boost network performance immediately. However, in bio-inspired routing protocols, acquiring a great network performance is a crucial and difficult challenge. The fact that no specific energy-efficient model was emphasized. In order to receive and transmit data packets, FFRP employs the basic energy-efficient model.

**Table 8.** Comparison of bio-inspired energy-efficient routing protocols (bio-inspired E2RPs).

| Protocol/ Year | Objective | Need of Localization | Implementation | Strategy of Energy Efficiency | Energy Efficiency | Advantages | Disadvantages |
|---|---|---|---|---|---|---|---|
| FFRP/ 2020 | Find reliable and stable routing | Yes | Simulation | Balancing communicating data packets traffic | High | Increased connection quality | Computational cost is high |
| MFPRP/ 2020 | Increased QoS | Yes | Simulation | Optimal route selecting for communicating data packet transmission | Low | Prevents sending duplicate data packets | Performance in terms of energy consumption is not better |

### 5.2. Cluster-Based Energy-Efficient Routing Protocols (C-b E2RPs)

Cluster-based (C-b) routing methods are among the most effective routing protocols for UWSNs. Sensor Rx nodes are grouped when employing C-b E2RPs, and each group has a cluster leader (CH). The CH gathers and mixes communicating data packets from the members of its cluster before transferring them from the Tx anchor node to the sensor Rx node. Since they manage most of the tasks and put in the greatest effort, however choosing a CH wisely is essential. UWSNs employs both clustering architectures, which are based on layer and grid configuration. (i) Using clustering-based protocols on layer, for which the seafloor is broken up into layers, with the sensor Rx nodes creating numerous clusters in each layer. The communicating data packets are gathered and combined by the cluster leader from its cluster members. The final communicating data packet is communicated to the next CH layer after communicating data packets from its cluster members when combined. Furthermore, in this sort of multi-hop mechanism used by UWSNs, communication data packets are sent from the Tx anchor node to the sensor Rx node. (ii) The concept of layer-based clustering routing protocols and grid-based clustering routing protocols are almost same. Layer-based clustering routing protocols split the seabed into various numbers of layers, whereas grid-based clustering routing methods divide the seafloor into various numbers of grids in UWSNs. This is the major distinction between the two types of clustering routing protocols. In this part, we reviewed E2RPs, which serve as the framework for UWSN clusters.

### 5.2.1. Energy-Efficient Routing Clustering Approach for UWSNs (E2RCA-UWSNs)

Despite the fact that, UWSN clustering is a well-known method. Clustering for UWSN energy-efficient routing protocols has just recently been deployed. In 2015, the first UWSN cluster routing protocol with improved energy efficiency was introduced. The E2RCA-UWSNs, in which a particular node serves as a CH, it was introduced by the author in [132]. Each individual node has a connection to the sensor Rx node; once these connections are made, the other communication nodes in each CH, when start to operate. The cluster members (CM) select the CH based on the network's shortest path between communicating nodes. Utilizing certain communicating nodes in this suggested routing strategy may aid to decrease the energy efficiency of UWSNs. However, it is very difficult and impossible to install these particular communicating nodes in this manner in UWSNs.

### 5.2.2. Energy-Efficient Grid-Based Clustering Routing Protocol (E2GRCP)

An energy-efficient clustered routing algorithm based on grid was presented by the author in [133] in 2016, based on the three-dimensional cube. In this routing system, the entire monitoring area was considered to be a cube, which was then further subdivided into several grids. In the E2GRCP approach, each grid made up a CH. A cube-length message sent by the base station informs the communicating nodes about the complete grid monitoring region. The communicating nodes set a timer and send a message to their close neighbors. They also give information about the grid area, the remaining energy, and the distance from the sink node. The sensor Rx node, which is located in the same grid region, assesses its remaining energy and the distance to the sink node while considering the message it has received. The time runs out if there is a significant amount of energy left in the message after it has been received from the Tx anchor node. However, if the message's remaining energy, which is received from the Tx anchor node, is low, it continues. The communicating node that has a high remaining energy and is near to the TX anchor node will thus be proposed as the CH. A multi-hop method is used to transmit data packets from the Tx anchor node to the CH. However, selecting the next Tx anchor node is a bit challenging. The distance to the sink node and the quantity of remaining energy are calculated by the Tx anchor node for the CH in which the cluster value based on the grid is less. The CH with the lowest weight value is selected as the subsequent forwarder node for data packet transmission. According to simulation results, the EECRC outperforms other clustered routing algorithms as VBF in [134], ERP2R in [135], EL-LEACH in [136], and LEACH in [137] in regards to energy efficiency.

### 5.2.3. Energy-Efficient Layer-Based Routing Protocol (E2LRP)

An energy-efficient routing protocol (E2LRP) based on layer for UWSNs was presented by the author in [138]. In a competition amongst the sensor Rx nodes to become a CH, the CH is chosen based on waiting time (WT), which is determined by the amount of energy left. A node becomes the CH and informs its nearby communicating nodes of this shift when the waiting period for a sensor Rx node ends. Clusters of communicating nodes located at greater depths in the sea have more communicating nodes per cluster than clusters located close to the water/ocean's surface. Therefore, it will have a negative impact on CH forwarding processes that are located close to the sink node. If the CH's remaining energy is less than the network's cluster average energy, the CH is considered as changed or damaged. The technique used to alter the CH helps to distribute the burden across the sensor Rx nodes. The use of flooding methods is expected to cause network congestion and considerable routing overheads.

### 5.2.4. Energy-Efficient Multi-Layer Cluster-Based Routing Protocol (MCE2RP)

The author of [139] introduced MCE2RP, an energy-efficient multi-layer routing protocol based on clusters, for UWSNs. According to the (MLCE2RP), the seabed is where the sink communicating nodes are located and they have infinite energy. Each communicating node in a UWSN determines the network's overall number of layers by using the

communicating node's depth. Every communicating node in the network has a network holding time, which is determined using the beginning energy and the remaining energy. A communicating node has a possibility to quickly become a CH in the network if it has a lot of leftover energy and has a short network holding time. Therefore, the communicating node transmits with its adjacent nodes when the network holding time expires. It is not feasible to become a CH if the adjacent node receives the information before the holding period for its own network expires. If two or more than two communicating nodes have the same network holding time, the CH is selected using the Bayesian spam filtering technique. The CH will be selected from among the communicating nodes with the highest probability in the network. The time division multiple access (TDMA) technology is used for the data packet transmission from the cluster member (CM) to the cluster head (CH). In this protocol, communicating nodes that penetrate the first network layer and send data packets directly to the sink node are not considered a part of the cluster. As a consequence of uneven load transfer among sensor Rx nodes that are closer to the network surface sink nodes, this assists in resolving the hotspot issue. However, if the network's CH updates are not considered, the CH's energy will soon run out and the network would finally come to an end.

### 5.2.5. Energy-Efficient Adaptive Clustering Algorithm (E2ACA)

For UWSNs, the author of [140] presented the energy efficient adaptive clustering algorithm (E2ACA). This technique can prevent a distant CH from the BS from prematurely expiring. In a sphere-shaped surveillance region, it employs a multilayer hierarchical technique. The BS is put in the middle of the application area. The number of network layers that make up the entire monitoring area varies, and each network layer is established in response to the CH radius' competition. This approach computes the width of each layer while also accounting for the CH's remaining energy.

Network communicating nodes with more energy remaining than the threshold value, might be suggested as CH candidates. The CH communicating node with the most weight among the CH candidates is ultimately selected as the network CH. The sum of the distance from the CH candidate to the nearby nodes, the average remaining energy of the neighbor nodes, the distance from the CH candidate to the BS, and the network maximum CH radius are used to determine the weight value. The data packet transmission is selected depending on the remaining CH energy and can handle both single-hop and multi-hop transmissions. If single-hop data packet transmission takes less energy than multi-hop data packet transmission, single-hop data packet transmission will be selected. Otherwise, adaptive clustering underwater network (ACUN) will be used as the multi-hop technique for data packet transmission. The simulation results in [140] demonstrate that, in regards to energy efficiency, single-hop data packet transmission performs better than multi-hop data packet transmission.

### 5.2.6. Energy-Efficient Grid-Based Clustering Routing Protocol (E2GCRP)

A clustering and energy efficient routing protocol based on the grid (E2GCRP) for UWSNs was proposed by the author in [141]. Each grid is created by using this technique as one cluster in the network. Based on the energy left over, the CH is placed there. The network's communicating nodes with the most remaining energy will transform into a CH. A coordinator communicating node, which is installed to assist with inter-cluster communication and data packet delivery to the BS, is a particular type of communicating node in the network. The technique cluster based underwater wireless sensor network (CUWSN) that has been proposed has the ability to boost network's throughput with the coordinating node's assistance. However, it is anticipated that the coordinator node and CH will terminate soon.

### 5.2.7. Comparison of Cluster-Based Energy-Efficient Routing Protocols (C-bE2RPs)

A thorough analysis of the E2RPs, which are based on a cluster for UWSNs, is shown in Table 9. By reducing the number of data packet transmissions, the E2RPs based on cluster are intended to conserve energy. Since energy efficiency is the major goal of all of the aforementioned routing protocols, the remaining energy of the sensor Rx nodes is an important factor for choosing the CH. However, the cluster center and the distance to the BS are key factors when choosing the CH. The E2RPs that are based on cluster are employed to improve network performance and reduced energy usage. However, the primary drawback of these E2RPs is their high network load and CH early termination, in extremely congested networks. There are several techniques that are suggested in [133,134,140,142–145] for UWSNs in regards to energy efficiency. All of these routing methods have the potential to significantly increase the load balancing of the network's communicating nodes, but they come at a higher computational cost. When deploying the communicating nodes, hierarchical or E2RPs based on layers may be employed to provide better and more effective cluster formation and good network performance. The aforementioned discussion indicates that many E2RPs are based on layer approaches that improve network performance. In inter cluster-based communication, multi-hop data packet forwarding becomes more frequent if all inter cluster-based communication is carried out via a single hop. Due to the close proximity of the communicating nodes, multi-hop inter cluster communication requires less energy than single-hop inter cluster communication. Furthermore, localization operation may appear difficult for the sensor Rx nodes in a dynamic UWSN environment, because it utilizes more hardware and energy. Therefore, the performance of localization-free techniques was excellent in terms of cost and energy efficiency.

**Table 9.** A thorough analysis of cluster-based energy-efficient routing protocols ((C-bE2RPs).

| Protocol/ Year | Position of a Node | Need of Localization | Change for New Cluster head | Energy Efficiency | Advantages | Disadvantages |
|---|---|---|---|---|---|---|
| E2RU-CA/ 2015 | Layer-based | No | No | High | Reduce energy consumption | End-to-end delay of the network is high |
| E2GRCP/ 2016 | Random and grid cube-based | Yes | Yes | Medium | Finds the shortest route to sink node | Overhead control is high |
| E2LRP/ 2018 | Layer-based | No | Yes | Low | Balancing the load | Overhead high routing and network congestion |
| E2ACA/ 2018 | Layer-based | Yes | No | Low | Load of first layer communicating nodes is very low | Control data packets exchange is high |
| MCE2RP/ 2019 | Layer-based | No | No | High | Load of first layer communicating nodes is low | Cluster head node is not changed during in communication |
| E2GCRP/ 2019 | Grid-based | Yes | No | High | Throughput performance of a repeater node is high | Early end of the cluster repeater node and the cluster head |

### 5.2.8. Cooperative Reliability-Based Energy-Efficient Routing Protocols (CO-RE2RPs)

Cooperative reliability-based E2RPs (CO-RE2RPs) based on reliability, which allows reliable data packet transmission from Tx anchor node to sensor Rx node, when considering the challenging underwater environment. Repeater nodes are used in these protocols to transmit data packets from the Tx anchor node to the sensor Rx node. The requirements of the network application heavily influence the choice of repeater nodes. This routing algorithm assists in boosting the network throughput by establishing a reliable connection between the sensor Rx node and the Tx anchor node. The same data packet is always received twice or more by the sensor Rx nodes, one from the Tx anchor node and the other from the repeater node. The sensor Rx node combines the data packets before extracting the

necessary information. This technique makes sure that another connection can assist with the successful delivery of data packets along those network channels that are impacted by unstable connections. However, these routing techniques do not address duplicate data packet transmissions, which improves the network's end-to-end latency. Figure 13, depicts the cooperative routing protocol for UWSNs.

### 5.3. Reliable Physical Distance-Based Energy-Efficient Routing Protocol (RPDE2RP)

A reliable E2RP (RPDE2RP), based on physical distance and remaining energy, was proposed by the author in [147]. RPDE2RP transmits data packets which are based on the remaining energy of the neighboring nodes and the quality of the link. The two phases of this protocol are (i) data packet transmission and (ii) cost development. Each sink node transmits a hello message during the second phase, where this message will be received by the sensor Rx node and utilized to determine the network cost. The physical distance of the network from every node remaining energy as well as the sink node will be utilized to determine the network cost. Furthermore, in the beginning phase, the Tx anchor node will transmit communicating data packets to network nodes whose cost value is no more than that of the Tx anchor node. The network's nearby nodes are displayed according to the remaining energies of the network. Any network node with a high level of remaining energy will be given the priority to become a Tx anchor node. Consequently, the priority will be given to a node to become a Tx anchor node if there are two or more nodes in the network that have the same and high remaining energy.

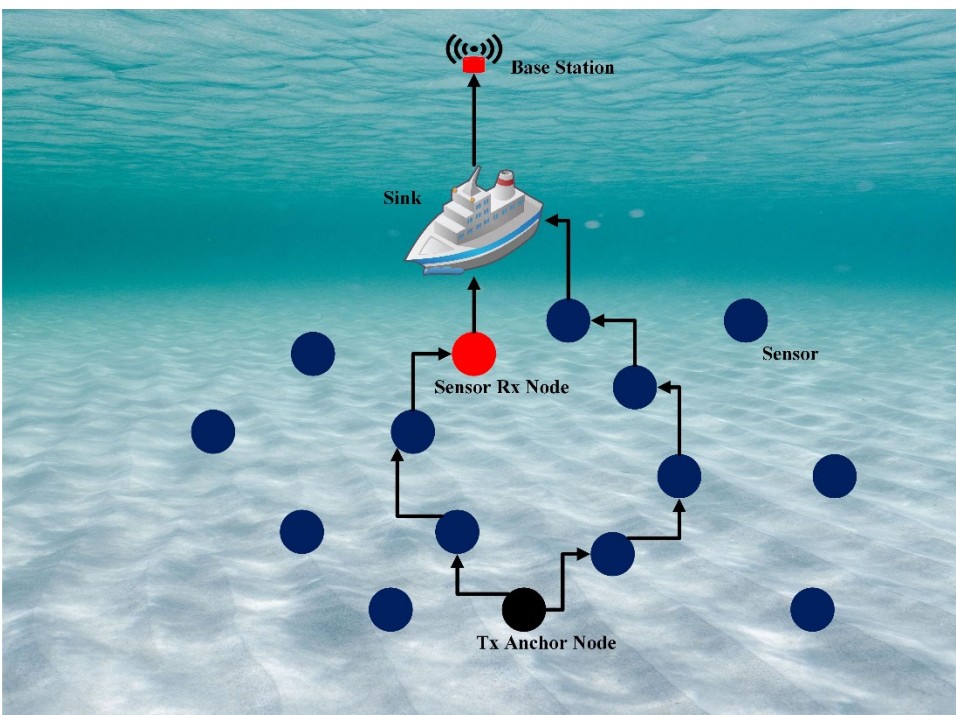

**Figure 13.** Cooperative reliability-based E2RPs (CO-RE2RPs) [146].

### 5.3.1. Reliable Energy-Efficient Routing Protocol (R-E2RP)

The author of [147] suggested a reliable E2RP for UWSNs, to reduce transmission delay by creating a path from the Tx anchor node to the sensor Rx node. RTS data packets are transmitted by the Tx anchor node to its nearby nodes. The nearby nodes responded with the delay of the data packet transmission to the Tx anchor node. The Tx anchor node chooses the subsequent hop as the one with the least data packet transmission delay by comparing the data packet latency issues. Each nearby node will receive a broadcast of the newly created path.

### 5.3.2. Cooperative Energy-Efficient Routing Protocol (CO-E2RP)

In [148] the author presented a cooperative and E2RP (CO-E2RP) for UWSNs. The repeater node is chosen from Tx anchor node to sensor Rx node, which is based on channel capacity and signal-to-noise ratio (SNR).

### 5.3.3. Energy-Efficient Multi-Path Grid-Based Geographic Routing Protocol (E2MG2RP)

An energy-efficient multi-route based on the grid geographic routing protocol (E2MG2RP) for UWSNs was presented by the author in [149]. A multi-route routing technique based on grid is utilized by the E2MG2RP. Every grid will use a network selection weight to determine with gateway node to use depending on the network's energy level and distance from the center of every grid cell. Similar to a repeater node, a gateway node's main objective is to transfer communicating data packets from one network grid to the next network grid. In the network the gateway node will be changed, if the presented gateway node remaining energy is low from the network threshold level. Furthermore, if a Tx anchor node has communicating data packets to transmit, it will send those communicating data packets to its neighbor gateway node. Therefore, the neighbor gateway nodes will select a right way to sensor Rx node by using the process of Round-Robin (RR) technique. The communicating data packets will be returned to the neighbor gateway node if there is no reliable route, and the neighbor gateway node will look for another suitable route to transmit these communicating data packets.

### 5.3.4. Energy-Efficient Cooperative Opportunistic Routing Protocol (E2CORP)

The energy efficient cooperative opportunistic routing protocol (E2CORP) for UWSNs was introduced by the author in [150]. The Tx anchor node first chooses a set of forwarding nodes before choosing the optimal repeater node from a group of nearby repeater nodes. In order to prevent data packets from being communicated through repeater nodes that were not selected for the forwarder set, the network employs a holding time. If the E2CORP achieves great performance in terms of energy efficiency, end-to-end delay of the network and communicating data packet delivery ratio. However, if the communicating nodes are located very far from one another, the repeater node selection technique in the network will perform poorly. Furthermore, when the communicating node positions are changed due to the ocean waves, then the sending of the communicating data packets to the chosen repeater nodes becomes a difficult task, which will cause more delays in the network.

### 5.3.5. Cooperative Energy-Efficient Routing Protocol (CO-E2RP)

In [151] the author presented a cooperative and energy efficient routing protocol (CO-E2RP) for UWSNs using both the data location and the sensor Rx nodes depth. In this routing protocol the sensor Rx node with the smallest location values and depth is chosen by the Tx anchor node. In terms of location values, the communicating node in the network which is closest to the sink node will have a smallest location value. As opposed to, the nearby node selected as a repeater node for forwarding communicating data packets, which is closest to sensor Rx node. However, this protocol increased the end-to-end delay of the network and unnecessary energy consumption occurs due to a lot of hearing.

### 5.3.6. Reliable Energy-Efficient Cross-Layer Routing Protocol (RE2CRP)

In [152] the author proposed a cross layer reliable energy efficient routing protocol (RE2CRP), to ensure the excellent data packet delivery in UWSNs. The (RE2CRP have two different phases: (i) Route change phase, (ii) Routing phase. During the first phase, every communicating node changes its self-routing information, with communicating node level, remaining energy, node ID, and distance. The received signal strength indicator (RSSI) value is used to calculate the distance from the nearby nodes to Tx anchor node. The next hop is chosen for data packet transmission using the routing table data.

### 5.3.7. Reliable Multi-Path Energy-Efficient Routing Protocol (RME2RP)

Similar to the aforementioned routing protocol (RE2CRP) in [152,153], where the author proposed another multi-route reliable and energy-efficient routing protocol (RME2RP) for UWSNs. The proposed routing protocol have also two phases: (i) Data packet forwarding phase, (ii) Path/route development phase. A special type of communicating node which is called a courier communicating node, where the sink node is directly connected, and deployed with each layer in the network. During the first phase of RME2RP, the regular sensor Rx nodes design a route from the Tx anchor node to the forwarding node and following the multi-path disconnected technique. The courier communicating nodes broadcast hello message, and after acquiring it, the adjacent communicating nodes join the multi-path disjoint technique and change its neighbor table. Furthermore, during the second phase of the RME2RP, the Tx anchor node broadcasts a path request through various links. When receive a path request, the routing table is updated by the nearby nodes. The path is selected on the basis of low cost of a link. However, in contrast with the aforementioned routing protocol RE2CRP in [152], the RME2RP uses the multi-path routing technique. Although the proposed protocol can achieve the reliability but cannot perform data packet redundancy.

### 5.3.8. Energy-Efficient Localization-Based Routing Protocol (E2LRP)

In [154] the author proposed an energy efficient routing protocol based on localization (E2LRP) for UWSNs. The proposed routing protocol uses three different kinds of beacon communicating nodes: (i) Unknown communicating nodes, (ii) Promoted beacon communicating nodes and (iii) Original communicating nodes. At three edge locations, a surface-based unknown communicating node can upgrade to a promoted beacon communicating node. The Tx anchor node follows normalized advancement connection matric model, to forwarding communicating data packets to sensor Rx node. Furthermore, using the normalized advancement matric model, a next forwarder node is select on the basis of the distance from Tx anchor node to sensor Rx node and remaining energy of the network.

### 5.3.9. Comparison of Cooperative Reliability-Based Energy-Efficient Routing Protocols (CO-RE2RPs)

Table 10 depicts, the basic characteristics of CO-RE2RPs for UWSNs. According to the aforementioned discussion, the main function of the energy efficient routing protocols are to supply a reliable communicating data packets and are also more energy efficient. These E2RPs can reduce the multiple route fading issues without utilizing various antennas. The energy saving approach is the main focus of the all-aforementioned energy efficient routing protocols with the selection of proper repeater nodes. Thus, the selection of repeater nodes in the network is necessary to increase the energy efficient communicating data packets transmission with very high reliability. A high network diversity gain can be generated by a reliable repeater node. In the network the best repeater nodes selection, theoretically improves the performance of the network and the main objective is cooperative routing, such as data packet delivery ratio, network throughput and reduce energy consumption. The overall network's concern is energy efficiency. However, a lot of the present routing protocols has used the remaining energy parameters, when selecting the repeater nodes in the network. As shown in, when choosing a repeater node in the network, both the remaining energy and the connection quality assessment are very important. Furthermore, the choice of many repeater nodes, as opposed to one, is the next popular route to ensuring the guaranteed communicating data packet delivery with low bit error rate. The main disadvantages of these routing protocols are that they compromise with network delay and ensuring communicating data packet reliability. Moreover, the communicating data packet duplication issue, that arises from multi-path routing leads to excessive energy use. Furthermore, in sparse networks the energy efficiency is limited. In this case, the communicating nodes must send communicating data packets over a considerable distance. Therefore, during the simulation results, a dense to sparse size of the network is understood

to demonstrate the protocol flexibility. As opposed to, if the expense is not a major obstacle, it is better to use several sink nodes than just using a single sink node.

**Table 10.** Comparison of the basic characteristics of CO-RE2RPs.

| Protocol/ Year | Selection of a Repeater Node | Optimality of Repeater Node Selection | Total Number of Repeater Nodes in Every Level | Control Data Packets | Mobility and Sink | Need of Lo-calization | Energy Effi-ciency | Advantages | Disadvantages |
|---|---|---|---|---|---|---|---|---|---|
| RE2RP/ 2014 | Distance from receiver node, link quality and remaining energy | Yes | Single | Yes | Static and multi | Yes | Low | End-to-end delay of the network is reduced | Control data packets are overhead |
| R-E2RP/ 2014 | Transmission delay of com-municating data packet | Yes | Single | Yes | Static and single | No | Medium | Communicating data packet sending is possible with minimum delay of transmission | When owing the lack of multi-route, the R-E2RP do not achieved full reliability of the network |
| COO-E2R PUWSN/ 2017 | Capacity of channel and SNR | Yes | Multiple | No | Static and multiple | Yes | Moderate | Data packet forwarding overlapping is none | Infrastructure cost is high |
| E2MG2RP/ 2016 | Remaining energy | Yes | Single | Yes | Static and single | Yes | High | Balancing the load between the sensor Rx nodes | The whole network is disturbed from the failure of the gateway node |
| E2COOORP/ 2017 | Fuzzy logic and fitness value | Yes | Multiple | No | Static and single | No | Moderate | Shortest path finding for routing | High delay, in sparse network the performance of E2COOORP is poor |
| COO-E2ORS/ 2018 | The distance from sensor Rx node is minimum | No | Single | Yes | Static and single | No | Medium | Data packet delivery ratio is increased | End-to-end delay of the network is high |
| RE2CRP/ 2018 | Distance between present and neighboring node, remaining energy, level of the node and neighboring nodes remaining energy | Yes | Single | Yes | Static and multi | No | High | Data packet redundancy is reduced | Due to overhearing no need of energy efficiency |
| RME2RP/ 2018 | Link quality and remaining energy | No | Multiple | Yes | Static and multi | No | High | Data packet delivery is reliable | Data packet delivery is redundant |
| E2LRP/ 2018 | Remaining energy and information of location | Yes | Single | Yes | Static and multi | Yes | Very High | E2LRP achieves better performance in terms of energy efficiency | Overhead is increased and the ratio of data packet delivery is low |

## 5.4. Reinforcement Learning Energy-Efficient Routing Protocols (RL-Based E2RPs)

Relationships between people and the outside world produce reinforcement learning (RL). How the intelligent agents behave in an unpredictably changing environment. An agent in RL accomplishes its objective, by interacting with and learning from its surroundings. To obtain a monetary incentive, RL gains worldly knowledge, what to do, and how to describe the conditions that led to the present conduct. Most of the time, the agent is not given instructions on what to do, therefore choosing the best course of action requires trial and error. According to the recent research studies, the Q-learning techniques are considered the most efficient methods in RL-based techniques. In Q-learning techniques, the agent decides on the basis of certain Q-value to improve the UWSNs lifetime.

There are several routing protocols that have been presented using the Q-learning techniques. In this paper, we have briefly discussed some routing protocols in terms RL-based energy efficiency.

### 5.4.1. Energy-Efficient Delay-Tolerant Q-Learning-Based Routing Protocol (E2DTQRP)

The first E2RP, which is based on the Q-learning technique for delay-tolerant underwater networks (E2DTRPQ), was presented by the author in [155]. A communicating data packet that is ready to be transmitted by a sensor Rx node characterizes the state of the network. Three factors are used to create the reward function: (i) Distance from relay node to sink node, (ii) Neighbor node density and remaining energy. According to the first criteria, a communicating node will be rewarded a high priority if it chooses a repeater node which is closest to the sink node. According to the second criteria, a communicating node will be rewarded on the basis of communicating node next layer and present layer density. A communicating node in the network, which selects a forwarder node will be awarded with a high priority, which resides in a high-density region that made easier forwarding process in the network. According the third criteria. A communicating node in the network that selects one specific communicating node, which have high remaining energy will be awarded a high communication priority. Furthermore, all communicating data packets are provided a deadline value, which is based on data packet priority. The communicating data packets are forwarded within the data packets deadline. Those data packets that are close to nearest deadline will be transmitted with most priority. The simulation results depict that, the suggested protocol achieves great performance in terms of energy efficiency and high data packet delivery.

### 5.4.2. Q-Learning-Based Energy-Efficient Lifetime Aware Routing Protocol (QE2LARP)

In [156], the author suggested a routing protocol based on Q-learning that is cognizant of network lifetime and is energy-efficient. In this protocol, the Q-values are determined on the basis of successful data packet transmission. In the Q-learning technique, an assistant's performance is determined by the reward value that is given based on their actions. The node distribution energy and the remaining energy of a node are used to formulate the proposed routing protocol reward function. The proposed protocol will select a route, in which a route has high remaining energy with shortest path. Therefore, if the protocol selects a route with low remaining energy, rather than negative reward will be offered. Although, when compared with vector-based forwarding (VBF) routing protocol, the proposed routing protocol achieves great performance in terms of energy efficiency and data packet delivery.

### 5.4.3. Q-learning-Based Energy-Efficient Routing Protocol (QL-E2RP)

The author of [157] proposes an E2RP for UWSNs based on the Q-learning technique named (QL-E2RP). The main aim of the proposed routing protocol is to provide and explore the efficient assets in hierarchically clustered networks. After the initial round of collecting communicating data packets, the Q-learning technique is continued after the BS is aware of the routing timeliness and energy consumption for processing the communicating data packets. The Q-value in this routing protocol is calculated, which is based on the transmission delay and remaining energy. In contrast, a regulatory element that balances the network's transmission delay and energy usage decides the reward. When the network regulatory element is set to zero, the aforementioned routing protocol only considers the residual energy. In addition, when the network regulatory element is set to one, then the focus of the aforementioned routing protocol is only to reduce the transmission delay of the network. According to [158] the proposed routing protocol achieves great performance in terms of network throughput, energy efficiency and also increases the lifetime of the network.

### 5.4.4. Comparison of Reinforcement Learning Energy-Efficient Routing Protocols

Table 11 shows, the comparison between the existing RL-based E2RPs. RL-based techniques are becoming more popular because they have high adaptability in underwater environment. Although the RL-based concepts had been used from many years. There are no routing protocols that are proposed on the basis of Q-learning techniques in terms of energy efficiency for UWSNs. In dynamic environment the energy efficient Q-learning techniques achieves great performance by using error and trial. Most energy efficient routing protocols based on Q-learning techniques are designed as Markov problems decisions. The network reward function, action space and state space architecture are challenging to utilize and relies entirely keeping in view the application objectives and requirements shows that, the majority of current E2RPs are considers in each individual data packet as a network state. Furthermore, the network reward function has a strong correlation with both the remaining energy and the distance from the repeater node to sink node. In comparison with other categories of E2RPs. E2RPs that are based on RL are still consider static and single sink nodes. Therefore, a large scope of research exists based on multiple sink node for routing protocols on the basis of energy efficient Q-learning techniques. The main disadvantages of the E2RPs based on Q-learning techniques are that they need to store and change its Q-value with each interaction, which makes the network denser. Considering this limitation, present E2RPs do not contemplate a huge network, and simulation results are performed only with less than 125 communicating nodes. Hence, the performance of E2RPs, which are based on Q-learning techniques are still questionable, since they need to be improved in terms of how they behave on large scale UWSNs.

**Table 11.** Comparison of RL-based E2RPs.

| Protocol/ Year | Objectives | State Space | Action Space | Reward | Need of Localization | Number of Sinks | Energy Efficiency | Advantages | Disadvantages |
|---|---|---|---|---|---|---|---|---|---|
| E2DTRPQ/ 2010 | Adaptability increased and reduced energy consumption | Individual Data packet | Forwarding data packet | Remaining energy and density of the node | No | Single | Moderate | Overhead control is reduced | The performance of E2DTRPQ in dense network is not suitable |
| QE2LARP/ 2010 | Network lifetime is increased with distributed remaining energy | Individual Data packet | Forwarding data packet | Probabilities of transmission and function value | No | Single | High | Increased the lifetime of the network | A lot of overhearing |
| QL-E2DRP/ 2019 | With the decrease of transmission delay increase lifetime of the network | Sensor Rx node position | Next Node | Distance of transmission | Yes | Single | High | Finds optimal route from Tx anchor node to sensor Rx node | Stability of link is not considered during in communication |

### 5.4.5. Depth-Based Energy-Efficient Routing Protocols (D-b-E2RPs)

A routing protocols based on depth are a typical routing technique for UWSNs. In this type of E2RPs, the forwarder communicating node is selected on the basis of the communicating node's depth. A hierarchical approach is used in the selecting the depth in network topology. When the depth level was less than the Tx anchor node present position. The Tx anchor node will select the forwarder node. This implies that, a communicating node which is near to the network sink node is the forwarder node, which was selected. In this part, E2RPs for UWSNs are discussed, which are based on depth.

### 5.4.6. Energy-Efficient Depth-Based Routing Protocol (E2D-bRP)

In [157] the author presented a free localization E2RP (E2D-bRP), which is based on depth. The E2D-bRP increases the lifetime of the network by decreasing the amount of network's transmissions. This proposed protocol is the enhanced version of depth-based routing protocol (D-bRP) discussed in [159]. There are two different phases of the whole communication scenario: (i), The phase of acquiring knowledge, (ii) The data packet

forwarding phase. In the first phase, the sensor Rx nodes communicate with their nearby nodes by broadcasting a hello message. It also includes their remaining energy levels and depth. Furthermore, during the second phase, using the data in the hello message, communicating, data packets are forwarded in the direction of the sink node. To reduce the amount of communicating data transmissions, out of its neighbors, the Tx anchor node chooses a group of forwarding nodes whose depth is less than its own. Since those communicating nodes which have lower depths are considered the neighboring nodes of the sink node.

For increased energy balancing between the communicating nodes in the network, a communicating node which have a large number of remaining energies could hold the communicating data packets for a smaller time to the comparison with the communicating nodes which have the remaining energy. The holding network time is calculated using a priority value to differentiate among the communicating nodes, which have the same level of energy and avoid multiple forwarding. Due to the hello message inclusion of the number of remaining energies, by choosing a group of communicating nodes for communicating data packet forwarding and avoiding redundant data packet transfer, E2D-bRP could minimize the energy consumption. E2D-bRP is extra efficient when compare with existing routing protocol DBR in [159], which are based on depth, in terms of end-to-end delay of the network and energy consumption. Efficient data packet transmission is needed for data about the nearby nodes, such as remaining energy and the depth, to be changed periodically. The lifetime of the network in E2D-bRP is increased to 40% which is greater than the existing routing protocol DBR for UWSNs.

### 5.4.7. Enhanced Energy-Efficient Depth-Based Routing Protocol (E3D-bRP)

An enhanced version of existing energy efficient routing protocol E2D-bRP in [157], and an enhanced energy efficient D-bRP (E3D-bRP) is presented in [160], which explains that how to increase the lifetime of the communicating nodes in the network, which lives in the middle depth level. Similar to existing E2D-bRP in [157], if a Tx anchor node have a communicating data packet, it will select a communicating node, which is very close to the network sink node for forwarding communicating data packet. A communicating node which has a high depth level value and high remaining energy, would have a chance to become a forwarder node. After receiving the communicating data packet, the network forwarding node will calculate the holding time on the basis of priority value and remaining energy. The value of priority helps the communicating nodes with dilemma, which arises in the network, when the energy level of different communicating nodes is same. In this scenario, a communicating node which have a high value of priority in the network will be selected as a repeater node. However, in contrast with the existing routing protocol E2D-bRP in [157], the E3D-bRP in [160] utilizes a technique of reactive routing to accommodate abrupt updates in the network.

### 5.4.8. Comparison of Depth-Based Energy-Efficient Routing Protocols (D-b E2RPs)

For routing protocol techniques in UWSNs, depth-based E2RPs are regarded as the foundational elements. All other E2RPs strategies used in UWSNs are built around this concept. The sensor Rx nodes in the network are deployed with respect to different depth levels in the undersea, with sea level depth being the most basic aspect of the underwater habitat. This crucial quality is considered by depth-based E2RPs, which are created based on depth in UWSNs. Although the routing techniques on the basis of depth are very old, Table 12 shows the current E2RPs which are based on depth. The first energy efficient routing protocol on the basis of depth was proposed in 2012, which was the enhanced DBR version that can handle communicating data packet redundant transmission. The main technique of both routing protocols is similar with each other. However, the main difference between these routing protocols is that the E2D-bRP can introduce new communicating nodes as an idle node in the network at the middle level of depth in UWSNs. When the communicating nodes loses their energy during the communication, the idle node supports

these normal nodes at the middle level of depth. The E3DBRP performance ID evaluated on the basis of different number of nodes, which are constant in the network.

**Table 12.** Comparison of based-depth energy-efficient routing protocols.

| Protocol/ Year | Control Data Packets | Need of Localiza- tion | Strategy of Energy Efficiency | Application Scope | Advantages | Disadvantages | Protocol/ Year | Control Data Packets | Need of Localiza- tion |
|---|---|---|---|---|---|---|---|---|---|
| E2DBRP/ 2012 | Yes | No | Forwarder node in the network is selected on the basis of remaining energy | Military surveillance and application of monitoring | Redundant communicating data packet transmission is reduced | Medium nodes of depth are early losing | E2DBRP/ 2012 | Yes | No |
| E3DBRP/ 2016 | Yes | No | A node in the network is selected as a forwarder node, which is closed to sink node and have a high remaining energy | Application scope of E3DBRP is time critical | During the transmission of medium nodes of depth, the network lifetime is increased | Throughput of the network is low | E3DBRP/ 2016 | Yes | No |

## 6. Current Problems and Research Difficulties

The current problems and research difficulties pertinent to constructing an E2RPs for UWSNs are taken into consideration in this section. In contrast to terrestrial networks, underwater data in UWSN systems incur significant bit error rates, multi-path dispersion, and propagation delays. Low dependability and excessive energy use during communicating data packet transmission are the results of these issues. In addition to these standard difficulties faced by UWSN networks, there are several unique problems that prohibits the lifetime of the network from being extended. These include issues with empty nodes, problem of hotspots, excessive high latency, limitation of ideal energy efficient route (E2R), limitation of privacy and security and unstable links. The difficulties and issues raised in this review should be useful to any interested research group.

### 6.1. Unstable Links

In UWSNs, the sensor nodes are moving all the time because of the sea waves. As a result, the routing connection is quite unreliable and the architecture of UWSNs seems to be very dynamic. Links that are unreliable frequently lose packets and have limited throughput, which significantly increases energy usage. In order to provide reliable communicating data packet forwarding, it is crucial to design a steady link quality.

### 6.2. Privacy and Security

One of the biggest issues for any sort of network, including UWSNs, is maintaining privacy and security. Due of their distant locations, applications of UWSN are quite vulnerable to malicious assaults. Without adequate security, the UWSN platform as a whole may become compromised, rendering all efforts useless. However, protection in routing, comes with higher energy expenses.

### 6.3. Problem of Hotspot

The network sensor nodes are often distributed in a hierarchy manner dependent on the water depth in most instances, where the UWSNs find utilization. Data is sent over many hops from the ocean floor to the leading base station. Communicating data packet transmission is a severe demand for the nodes located near to the surface. Therefore, the devices of sensor nodes at the base station are more probable to run outside from power than the communicating nodes on the lowest level. To lessen the burden of data packet transformation, one important strategy for overcoming this issue is to stop the sensor nodes near to the base station from creating clusters. However, the early mortality issue with near the surface layer sensor nodes still remains, and cannot be totally avoided. Therefore, in order to extend the lifespan of the network, the problem of hotspot needs to be addressed.

*6.4. QoS with Routing*

Many of the existing QoS-enabled routing techniques are restricted to certain applications and take just a few metrics under consideration. A lack of harmony exists between maintaining QoS and preserving energy. In this sense, future research in several applications or multiple UWSNs should focus on energy-efficient routing with QoS.

## 7. Conclusions

Due to the dynamic, complicated, and harsh characteristics of the underwater habitat, UWSNs faces various difficulties. Despite these challenges, one of the most critical issues to address for IoUT that can be used generally is energy efficiency/energy consumption. During the previous couple of decades, a lot of research efforts have been directed to dealing with these challenges of energy efficiency/energy consumption in UWSNs. After doing a comprehensive survey we found out that our study can be divided in to two subcategories, i.e., energy-efficient MAC and routing protocols for UWSNs. We compared different MAC and routing protocols that are used for energy efficiency by different research groups, based on their design. It is concluded from the survey that a homogenous ideal solution in terms of energy efficiency for MAC and routing protocol cannot exist for UWSNs, due to the changing need according to the change in communication environment. However, we find out from this study that the performance analysis of Buffering_slotted_Aloha is best from the other aforementioned methods opted for UWSNs, because after the comparison with other ALOHA protocols the Buffering_slotted_Aloha protocol achieves significant performance in terms of average delay, network throughput ratio and energy efficiency/energy consumption. Furthermore, Artificial intelligence-based routing solutions have gained popularity because of their capacity for adaptation in a dynamic environment and their ability to satisfy a variety of application needs. The existing energy-efficient MAC and routing strategies for every class were also compared and summarized. Compared to other categories, routing protocols based on cooperative reliability, cluster, and MAC protocols based on ALOHA have received the most research in this article. Finally, the issues that are still needed to be resolved and the challenges in conducting research on E2RPs in UWSNs are emphasized as directions for future study.

**Author Contributions:** Conceptualization: Q.G. and Z.U.K.; methodology: Q.G. and Z.U.K.; software: Z.U.K.; validation: Q.G.; formal analysis: A.M., Z.U.K. and M.M.; investigation: Q.G.; resources: Q.G. and Z.U.K.; data curation: A.M., M.E.A. and M.M.; writing—original draft preparation: Z.U.K.; writing—review and editing: A.M., S.U.K., J.K. and M.M.; supervision: Q.G.; project administration: M.E.A.; funding acquisition: M.E.A., G.A. and I.U. All authors have read and agreed to the published version of the manuscript.

**Funding:** This work is supported by the College of Underwater Acoustics Engineering, Harbin Engineering University, Heilongjiang, Harbin, China, National Natural Science Foundation of China (NSFC) under grant no U1806201.

**Informed Consent Statement:** Not Applicable.

**Data Availability Statement:** The data presented in this study will be available on request from the corresponding author.

**Acknowledgments:** This work was supported by the EIAS Data Science and Blockchain Lab, College of Computer and Information Sciences, Prince Sultan University, Riyadh Saudi Arabia.

**Conflicts of Interest:** It is declared that authors have no conflict of interest.

## Abbreviations

| | | | |
|---|---|---|---|
| WSN | Wireless sensor networks | ALOHA-CA | ALOHA with carrier sense |
| UWSNs | Underwater wireless sensor networks | ALOHA-CS | ALOHA with collision avoidance |
| TWSNs | Terrestrial wireless sensor networks | EI-ALOHA | Equal interval ALOHA |
| UWASNs | Underwater acoustic sensor networks | L-ALOHA | Learning ALOHA |
| UW | Underwater | P-ALOHA | Pure ALOHA |
| MAC | Media access control | S-ALOHA | Slotted ALOHA |
| ACK | Acknowledgment | Slotted-CS-ALOHA | Slotted carrier sense ALOHA |
| CS | Carrier sense | ST-Slotted-CS-ALOHA | Saving time slotted carrier sense ALOHA |
| CTS | Clear-to-send | VI-ALOHA | Variable interval ALOHA |
| RTS | Request-to-send | T-LOHI | Tone-LOHI |
| CDMA | Code division multiple access | OP | Operating system |
| TDMA | Time division multiple access | E2RPs | Energy-efficient routing protocols |
| FDMA | Frequency division multiple access | FFRP | Firefly mating optimization routing protocol |
| CSMA | Carrier sense multiple access | MFPRP | Memetic flower and energy-efficient pollination routing protocol |
| ERCA-MAC | Energy-efficient reliable and cluster-base adaptive MAC | C-b | Cluster-based |
| H-MAC | Hybrid-MAC | CH | Cluster head |
| KHZ | Kilo-hertz | CM | Cluster members |
| MACA | Multiple access collision avoidance | E2GRCP | Energy-efficient grid-based cube routing protocol |
| PN | Pseudo noise | E2LRP | Energy-efficient layer-based routing protocol |
| RF | Radio frequency | MCE2RP | Energy-efficient cluster-based multi-layer routing protocol |
| P-MAC | Preamble MAC | E2ACA | Energy-efficient adaptive clustering algorithm |
| UAMC-MAC | Underwater acoustic multi-channel MAC | E2GCRP | Energy-efficient grid-based clustering routing protocol |
| UW Sink | Underwater sink | CO-RE2RPs | Cooperative reliability-based energy-efficient routing protocols |
| ER-MAC | Efficiency reservation MAC | RPDE2RP | Reliable physical distance-based energy-efficient Routing protocol |
| GC-MAC | Graph coloring MAC | CO-E2RP | Cooperative and energy-efficient routing protocol |
| DL-MAC | Depth Layering MAC | E2MG2RP | Energy-Efficient Multi-Route Grid-based Geographic Routing Protocol |
| LO-MAC | Latency-Optimized MAC | E2CORP | Energy Efficient Cooperative Opportunistic Routing Protocol |
| ST-MAC | Spatial-Temporal MAC | RE2CRP | Reliable Energy-Efficient Cross-Layer Routing Protocol |
| FF-MAC | Fitness-Function-based MAC | RME2RP | Reliable Multi-Route Energy-Efficient Routing Protocol |
| CSMA/CA | Carrier-Sense-Multiple-Access/Collision-Avoidance | E2LRP | Energy-Efficient Localization-based Routing Protocol |
| ALOHA-AN | ALOHA with Advance-Notification | RL | Reinforcement Learning |

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
