# Peer review of "A Comprehensive Survey of Energy-Efficient MAC and Routing Protocols for Underwater Wireless Sensor Networks"

_electronics, doi:10.3390/electronics11193015_

Round 1
Reviewer 1 Report
-The authors studied the two categories, i.e. Energy-Efficient MAC and routing protocols in UWSNs. They compared different MAC and routing protocols by different research groups, based on their design. They find out that a homogenous ideal solution in terms of energy efficiency for MAC and routing protocol cannot exist for UWSNs, due to the changing need according to the change in communication environment.
-The following major corrections are required:
-The introduction section is too general, and it introduces concepts that are well known to the energy-efficient MAC and routing protocols in UWSNs. The introduction does not stimulate to go ahead with the remaining of the paper because it does not introduce any really new topic/solution. Furthermore, “the motivation and research challenges” at the introduction section is missing. Please rewrite this section.
-Please provide a section named “Research methodology” as a guideline for searching relevant articles in the Energy-Efficient MAC and routing protocols in UWSNs including research questions, search keywords, article selection,…
-Your study is very similar to the following papers. What is the difference between your study and mentioned papers?
https://www.sciencedirect.com/science/article/abs/pii/S1084804516301230
https://ieeexplore.ieee.org/abstract/document/9398696
-This survey paper is mainly based on the classification of existing energy-efficient MAC protocols into three categories: frequency domain, full bandwidth and hybrid. However, no motivation and/or justification are provided that explains why such a categorization is important and how this will help/facilitate the researchers of the field. Furthermore, the description in Section 4 that briefly explains each category, is mostly vague and not sufficient. This description should be clearer and more theoretical as each category represents a different set of problems associated with the field of energy-efficient MAC protocols. E.g., what are the similarities and differences between frequency domain and full bandwidth at the energy-efficient MAC and routing protocols.
Lastly, some key papers in the research area are left out and question and require improving the search methodology. Some of the related papers that should have be included are:
https://link.springer.com/article/10.1007/s11276-022-03061-2
https://www.mdpi.com/2077-1312/10/4/488
https://onlinelibrary.wiley.com/doi/abs/10.1002/spe.2641
https://ieeexplore.ieee.org/abstract/document/9760848
https://link.springer.com/article/10.1007/s00500-020-05409-2
https://www.mdpi.com/2079-9292/11/16/2590
…
-Paper needs some revision in English. The overall paper should be carefully revised with focus on the language: especially grammar and punctuation.
-Overall, there are still some major parts that the authors did not explain clearly. Some additional evaluations are expected to be in the manuscript as well. As a result, I am going to suggest Major revision the paper in its present form.
Reviewer 2 Report
This manuscript reviews energy-efficient MAC and routing protocols for underwater wireless sensor networks, which has the following problems:
1. There are many places in the manuscript where references are misquoted (e.g. 137, 139).
2. In Chapter 4, the number of titles was divided into too many levels. Whether you can combine several sections or change the lower title number to another symbol.
3. The last two sections of chapter 4 (558 and 568) have incorrect title numbers. The serial numbers of these two sections are 4.3.2.2.7 and 4.3.2.2.8 respectively, but the serial number of the previous section is 4.3.2.2.12.
Reviewer 3 Report
The authors have reviewed the progress of the medium access control and routing protocols for underwater wireless sensor networks, including the improvement in the performance in terms of throughput and end-to-end delay, as well as the design of the energy efficient MAC and routing protocols. Analyses and discussions are interesting and helpful for next-generation underwater wireless sensor networks. This paper can be published in Electronics, provided following issues can be addressed.
1. Some abbreviations should be clarified when they appear for the first time.
2. The language can be double-checked and improved.
3. In Page 17, check the indices of Figure 1, Figure 2, Figure 3.
4. Add more discussion regarding the practical applications of the discussed approaches in other sensing systems
See e.g.
Z Wang et al., Ultra-sensitive DNAzyme-based optofluidic biosensor with liquid crystal-Au nanoparticle hybrid amplification for molecular detection, Sensors and Actuators B: Chemical, 2022.
M Lee et al., Wearable wireless biosensor technology for monitoring cattle: a review, Animals, 2021.
Reviewer 4 Report
The author should accurately address the below comments.
- Abstract: The abstract is long and requires reducing the number of words without losing scientific meaning. For the abstract to be comprehensive, we recommend adding the findings from this review.
- Keywords: We suggest that the authors should replace keywords such as “Routing Protocols” and “Underwater wireless sensor networks” because these keywords are already found in the article title. It is better that they replace them with other keywords to increase the reach of the article.
- Introduction Section: Why were the bullet points (lines94-130, pages2-4) added suddenly? The main contributions at the end of the Introduction section are not organized.
- Why did this research not address the shortcomings of previous reviews/surveys on the subject?
- The title of the section "Most important dares of the (UWSNs) transmission" is unclear.
- Are Tables 1 and 2 belong to the authors? If so, then there is no problem. If they pertain to other research, references should be made to these references.
- Data aggregation in UWSN is not described in detail.
- We believe that the structure of the methodology is not written sequentially.
- Acronyms must define before using in the first appearance such as CSMA … etc.
- There are some unnecessary sentences that require removal.
- Figures and Tables: Some figures should be drawn with high resolution such as 1, 3, 4, 5 … etc All figures and tables are not used in the text. Figure 6 requires miniaturization. Also, some figures do not have references.
English Writing: This paper requires moderate proofreading. The author should sift through the entire paper to remove all grammar and typographical problems in terms of English writing. Also, authors should standardize English writing (either American or British). Some paragraphs are very long and should be broken down to be clearer.
Round 2
Reviewer 1 Report
Thanks to authors for the detailed response and additions I read through the comments and skimmed the revised PDF, The updates did improve the paper a lot. I would be happy to recommend this paper for publication |
Reviewer 4 Report
The authors did not respond to some comments. The author should accurately address the below comments.
- Introduction Section: Why were the bullet points (lines94-130, pages2-4) added suddenly? This comment has not been responded to accurately.
- Why did this research not address the shortcomings of previous reviews/surveys on the subject? This comment is not addressed. The authors should add some reviews papers, highlight its flaws, and demonstrate the superiority of this review over previous (not research papers) reviews.
- Data aggregation in UWSN is not described in detail. This section still needs more details about how the UWSN collect data.
- Acronyms must define before using in the first appearance such as CSMA … etc.
- We believe that the structure of the methodology is not written sequentially. This section still requires improvement.
- Figures and Tables: There still are figures and tables are not used in the text. Figures and tables are not used in order in the text. Also, some figures do not have references.
- English Writing: This paper still requires moderate proofreading. The author should sift through the entire paper to remove all grammar and typographical problems in terms of English writing. Also, authors should standardize English writing (either American or British) such as “analyze" and “analyse”. Some paragraphs are very long and should be broken down to be clearer.
Round 3
Reviewer 4 Report
The authors did not respond to some concerns. The authors should accurately address the below comments.
- The authors did not address our comment “Why did this research not address the shortcomings of previous reviews/surveys on the subject? This comment is not addressed. The authors should add some reviews papers, highlight its flaws, and demonstrate the superiority of this review over previous (not research papers) reviews.” accurately, they added references [35], [36], [37] and [38] and this is good but they did not criticize the included surveys.
- Acronyms must define before using in the first appearance such as CSMA … etc. Please address our comment accurately.
- We believe that the structure of the methodology is not written sequentially. This section still requires improvement. The authors' answer did not convince us. All types of research should include a specific methodology.
- Figures and Tables: Figures and tables are not used in order in the text. Figures 2, 7 …etc. are not used in-text. Also, some figures do not have references.
English Writing: The authors did not address this comment. This paper still requires moderate proofreading. The author should sift through the entire paper to remove all grammar and typographical problems in terms of English writing. Also, authors should standardize English writing (either American or British) such as “analyze" and “analyse”. Some paragraphs are very long and should be broken down to be clearer.
